



Hydrology and
Earth System
Sciences

# Seasonal hydro-meteorological forecasts for Europe: sources of skill TS1

**Wouter Greuell, Wietse H. P. Franssen, and Ronald W. A. Hutjes**

Water Systems and Global Change (WSG) group, Wageningen University and Research, Droevendaalsesteeg 3,
6708 PB Wageningen, the Netherlands TS2

**Correspondence:** Wouter Greuell (wouter.greuell@wur.nl)

**Abstract.** TS3 This paper uses hindcasts (1981–2010) to investigate the sources of skill in seasonal hydrological forecasts for Europe. The hindcasts were produced with WUSHP (Wageningen University Seamless Hydrological Prediction system). Skill was identified in a companion paper. In WUSHP, hydrological processes are simulated by running the Variable Infiltration Capacity (VIC) hydrological model forced with an ensemble of bias-corrected output from the seasonal forecast system 4 (S4) of the European Centre for Medium-Range Weather Forecasts (ECMWF) CE1. We first analysed the meteorological forcing. The precipitation forecasts contain considerable skill for the first lead month but hardly any significant skill at longer lead times. Seasonal forecasts of temperature have more skill. Skill in summer temperature is related to climate change and is more or less independent of lead time. Skill in February and March is unrelated to climate change. Different sources of skill in hydrometeorological variables were isolated with a suite of specific hydrological hindcasts akin to ensemble streamflow prediction (ESP). These hindcasts show that in Europe, initial conditions of soil moisture (SM) form the dominant source of skill in run-off CE2. From April to July, initial conditions of snow contribute significantly to the skill. Some remarkable skill features are due to indirect effects, i.e. skill due to forcing or initial conditions of snow and soil moisture at an earlier stage is stored in the hydrological state (snow and/or soil moisture) of a later stage, which then contributes to persistence of skill. Skill in evapotranspiration (ET) originates mostly in the meteorological forcing. For run-off we also compared the full hindcasts (with S4 forcing) with two types of ESP (or ESP-like) hindcasts (with identical forcing for all years). Beyond the second lead month, the full hindcasts are less skilful than the ESP (or ESP-like) hindcasts, because inter-annual variations in the S4 forcing consist mainly of noise which enhances degradation of the skill.

## 1 Introduction

Society may benefit from seasonal hydrological forecasts (Viel et al., 2016; Soares and Dessai, 2016; Crochemore et al., 2016), i.e. hydrological forecasts for future time periods from more than 2 weeks up to about a year (Doblas-Reyes et al., 2013). Such predictions can be exploited to optimise, for example, hydropower energy generation (Hamlet et al., 2002), navigability of rivers in low flow conditions (Li et al., 2008) and irrigation management (Ghile and Schulze, 2008; Mushtaq et al., 2012) to decrease crop yield losses.

This is the second paper about seasonal hydrological forecasts for Europe produced with WUSHP (Wageningen University Seamless Hydrological Prediction system), a dynamical (i.e. model-based) system. In summary, the forecasts of WUSHP are made with the Variable Infiltration Capacity (VIC) hydrological model, which uses bias-corrected output of forecasts from the seasonal forecasting system 4 (S4) of the European Centre for Medium-Range Weather Forecasts (ECMWF) as meteorological forcing. The system is probabilistic.

In the present and in the companion paper (Greuell et al., 2018), WUSHP is used as a research tool for purposes of academic interest. In the companion paper, the set-up of WUSHP has been described, and spatial and temporal variations of skill, or the lack thereof, in run-off and discharge in Europe have been established by means of hindcasts. Sig-

nificant skill was found for many regions, varying by initialisation and target months. For lead month 2, hotspots of significant skill in run-off are situated in Fennoscandia (for target months from January to October), the southern part of the Mediterranean (from June to August), Poland, northern Germany, Romania and Bulgaria (mainly from November to January) and western France (from December to May). In general, the spatial pattern of significant skill in run-off was found to be fixed in space, while the skill decreased in magnitude with increasing lead time. Some significant skill remained even at the end of the hindcasts (7 months).

To extend the evaluation of the system, its reliability was analysed. The main finding is that during the 2 first lead months the system is not far from being perfectly reliable but that with progressing lead time reliability is reduced. We also found that discrimination skill and reliability have similar characteristics, e.g. for longer lead times the highest values of reliability are found in some regions with considerable amounts of discrimination skill. Details of this analysis are provided in Appendix A.

The current paper aims to identify the sources of the skill in WUSHP and is structured in two main parts. In the first part, an analysis of the skill in the most important meteorological forcing variables (precipitation, 2 m temperature and incoming short-wave radiation from S4) is presented. For S4, this was done earlier by Kim et al. (2012) for the boreal winter months (DJF), with initialisation on the first of November. For that case, they found that in Europe, S4 has no skill in the precipitation forecasts and some skill in the temperature forecasts for southern Sweden, southern Finland, the region southeast of Saint Petersburg and northern Germany. Scaife et al. (2014) analysed the skill for the same target months and starting date but with another prediction system, namely the Met Office Global Seasonal Forecast System version 5 (GloSea5). They found that, while the GloSea5 temperature forecasts for Europe contain hardly any significant skill, the GloSea5 forecasts of the North Atlantic Oscillation are correlated significantly with observed temperatures in northern and southern Europe. This means that there is untapped predictability in the GloSea5 temperature forecasts. We will analyse predictability of the mentioned output variables of S4 for the whole continent and will consider all combinations of lead and target months.

The second line of analysis aims to investigate the reasons for the presence or absence of skill in hydro-meteorological variables by means of a series of specific hindcasts that isolate potential sources of skill, namely meteorological forcing, the initial conditions of soil moisture (SM) and the initial conditions of snow. Such an approach was explored earlier by Wood et al. (2005), Bierkens and Van Beek (2009) [TS4] and Koster et al. (2010). Each specific hindcast is basically identical to the standard hindcasts that we analysed in the companion paper, named full streamflow hindcasts (FullSHs [CE3]; climate-model-based hindcasts" according to Yuan et al., 2015). However, in the specific hindcasts, one or two

of the sources of predictability are isolated by eliminating the effect of all of the other sources through the removal of their inter-annual variation. In the ensuing analysis the skills in hydro-meteorological variables found in the different specific hindcasts will then be compared among themselves and with the skill from the FullSH.

These specific hindcasts are similar in structure to and inspired by the conventional ensemble streamflow prediction (ESP) technique (e.g. Wood and Lettenmaier, 2008; Shukla and Lettenmaier, 2011; Singla et al., 2012), which can, like our specific hindcasts, be used to isolate sources of skill. The main difference between the specific hindcasts of this study and the ESP technique is that in ESP and its variant reverse ESP, the meteorological forcing is taken from data based on observations, while in the present study the forcing is taken from meteorological hindcasts. In fact, we also produced ESP. In Sect. 4.3 we will compare these with one of the other specific hindcasts and discuss the relation between our specific hindcasts and the ESP suite more generally.

Though this paper focusses on run-off, the analysis is complemented with an analysis of the skill in evapotranspiration (ET), since this variable has a large effect on run-off (see Willmott et al., 1985). Predictions of evapotranspiration also have independent value, because they are useful for planning of water level control in polders and for planning of water use for irrigation and fertiliser application. As for run-off, we will exploit the specific hindcasts to isolate the different sources of predictability in evapotranspiration forecasts.

The version of VIC that we used was only crudely calibrated (Nijssen et al., 2001). Hence, discharge computed by the present version of the system may be expected to deviate substantially from observations, both in terms of the mean and in terms of the spread of the ensemble of forecasts. Also, within WUSHP no post-processing of discharge is carried out to correct for such deficiencies. This makes the system unsuitable for issuing forecasts of absolute amounts of discharge, but the system can be used to provide information on how likely it is that in a coming month or for season discharge, it [CE4] will be above or below normal. Consequently, the most important criteria for the selection of skill metrics (see Sect. 2.2) are their ability of discrimination and their insensitivity to biases and to the spread of the forecasts.

The objective of the present paper is to analyse, at a pan-European and regional scale, the sources of probabilistic skill of seasonal hydrological forecasts produced by WUSHP. The next section (Sect. 2) will describe the seasonal prediction system itself and the analysis approach as well as details of the various specific hindcast performed. We will present the skill in the meteorological forcing (Sect. 3.1), isolate the skill in run-off due to either forcing or different types of initial conditions (Sect. 3.2), and finally analyse the skill in evapotranspiration (Sect. 3.3). We conclude with a discussion (Sect. 4) and conclusions (Sect. 5).

## 2 System and methods

### 2.1 The forecast system

The forecasts of WUSHP combine three elements, namely meteorological forcing from ECMWF's Seasonal Forecast System 4 (Molteni et al., 2011), bias correction of the meteorological forcing with the quantile mapping method of Themeßl et al. (2011) and simulations with the VIC hydrological model (Liang at al., 1994). The skill of the system was assessed with hindcasts. These cover the period 1981–2010, were initialised on the first day of each month and extend to a lead time of 7 months. The system is probabilistic (15 members), so each set of hindcasts consists of a total of 5400 runs (30 years $\times$ 12 months $\times$ 15 members TS5). In addition a single reference simulation was performed in which VIC was run with a gridded data set of model-assimilated meteorological observations, namely the WATCH Forcing Data Era-Interim (WFDEI; Weedon et al., 2014). The reference simulation has a dual aim. The first aim is to create initialisation states for the hindcasts. Secondly, the output of the reference simulation, e.g. run-off, is used for verification of the hindcasts. This output will be named "pseudo-observations" here.

Due to the set-up of the routing module of VIC, the state of discharge could not be saved and loaded. Hence for spin-up discharge, each 7-month hindcast was preceded by a 1-month simulation with WFDEI forcing, which in turn was initialised with the model states generated in the reference simulation and zero discharge. All hindcasts and simulations were performed on a $0.5° \times 0.5°$ grid in natural flow mode, i.e. river regulation, irrigation and other anthropogenic influences were not considered. VIC is run with a time step of 3 h. More details about the set-up of the system and the hindcasts can be found in the companion paper (Greuell et al., 2018).

### 2.2 Methods of analysis and observations

In this paper we analyse hindcasts of run-off, discharge and evapotranspiration. Run-off is defined as the amount of water leaving the model soil either, along the surface or at the bottom, while we define discharge as the flow of water through the largest river in each grid cell.

Discrimination skill (briefly skill from now on) is measured in terms of the correlation coefficient between the median of the hindcasts and the observations (or pseudo-observations; $R$). We will designate $R$ values as significant for $p$ values less than 0.05. We also considered metrics designed for the evaluation of categorical forecasts (terciles), namely the relative operating characteristics (ROC) area and the ranked probability skill score (RPSS). The thresholds used for assigning individual (pseudo-)observations to terciles were determined from the (pseudo-)observations themselves. Similarly hindcasts were assigned to terciles by reference to themselves. Due to this strategy metrics are unaffected by biases, a desired property (see Sect. 1). In the com-

panion paper skills in terms of the considered metrics were compared, and it was found that for all combinations of target and lead months the skill patterns in the maps were similar to a high degree. For that reason, we selected only one of them ($R$) for this paper.

Unless mentioned otherwise, prediction skill of the hydrological variables is determined against the pseudo-observations (see Sect. 2.1). These have the advantages of being complete in the spatial and the temporal domain and of being available for all model variables. We will refer to this type of skill as "theoretical skill". In the companion paper theoretical skill for discharge was compared to "actual skill", which is the skill assessed with real observations. For the determination of the skill of the meteorological forcing we used the WFDEI data.

To investigate the possible contribution of trends to skill, skill in the meteorological forcing and in run-off was determined both before and after removing the trend from both the (pseudo-) observations and the hindcasts. Data were detrended by first constructing time series (1981–2010) for each variable, target month, lead month and grid cell (30 values). We then removed the trend from each time series by first fitting a least-squares regression line to the time series and then subtracting the time series corresponding to the line from the original data. For the hindcasts, time series were constructed for the mean of the ensembles, and the resulting best fit was subtracted from each member individually.

Like in the companion paper, skill was analysed on a monthly and not on a seasonal basis with the aim of achieving a relatively high temporal resolution in the skill analysis. Attention was confined to consistent skill, which we define as skill that persists during at least two consecutive target or lead months. In accordance with Hagedorn et al. (2005), we designated the first month of the hindcasts as lead month zero.

In most result sections, we will first analyse and explain skill at the level of the entire domain. We will then take out the most noteworthy details of the summary plots and seek an explanation for them.

### 2.3 Isolation of sources of skill and surface water initialisation

As already pointed out in the introduction, a number of specific hindcasts were carried out with the aim of isolating the contributions of different sources to skill. The FullSHs, in which skill is due to both meteorological forcing and initial conditions, constitute the starting point. The specific hindcasts can be seen as restricted, in the sense of limiting the types of sources of skill, versions of the FullSH. The following five sets of specific hindcasts, each consisting of 5400 computer runs, were produced:

1. The *InitSHs* isolate the skill due to both types of initial conditions considered here (soil moisture and snow). Like in the FullSH, the annually varying initial con-

ditions are taken from the reference simulation, while for each year the meteorological forcing is identical and consists of an ensemble of 15 S4 hindcasts. More specifically, we selected member 1 from the 1981 hindcasts, member 2 from the 1983 hindcasts, etc. By using identical meteorological forcing for all of the years of the hindcasts, skill in hydro-meteorological variables due to skill in the forcing is eliminated.

2. The *SMInitSHs* isolate the skill due to the initial conditions of soil moisture only. The SMInitSH is identical to the InitSH, but in all SMInitSHs snow initial conditions are taken as the 30-year average of the snow conditions in the reference simulation.

3. The *SnInitSHs* isolate the skill due to the initial conditions of snow contained in the snow cover. The SnInitSH is identical to the InitSH, but in all SnInitSHs soil moisture initial conditions are taken as the 30-year average of the soil moisture conditions in the reference simulation.

4. The *MeteoSHs* isolate the skill due the meteorological forcing and as such are the full complement of the InitSH. Like in the FullSH, the annually varying forcing is taken from the probabilistic S4 hindcasts, while for each year the initial soil moisture and snow conditions are identical and equal to the 30-year average of the soil moisture and snow conditions in the reference simulation. By taking identical initial conditions for all of the years of the hindcasts, skill due to the initial conditions of soil moisture and snow is eliminated.

5. The *ESP* are identical to the InitSH, both in terms of their construction and in terms of their purpose. However, in the ESP the forcing is not taken from the S4 hindcasts but from the WFDEI data by selecting the 15 odd years from 1981 to 2009.

Forcings and initial conditions of all of these hindcasts differ among the calendar months so that the annual cycle is conserved. Hence, in the list above, the following apply:

– "Identical for all years" means that the forcings (or the initial conditions) for all hindcasts starting in, for example, May are identical.

– "30-year average" means that the initial conditions for all hindcasts starting in, for example, May are averaged over all of the 1 May model states in the reference simulation.

– "Annually varying" means that the forcings (or the initial conditions) for all hindcasts starting in, for example, May vary from year to year.

These statements also hold for the other calendar months.

Thus, like the FullSH, all specific hindcasts for a single starting date consist of 15 members, which is important, since ensemble size affects skill metrics (Richardson, 2001). Also, in all hindcasts the probabilistic character is exclusively due to the 15 members of the meteorological forcing, while initial conditions are deterministic. This consistency is important, since the main aim of the various specific hindcasts is to compare them with each other. A disadvantage of the small ensemble size is the sampling uncertainty (see Sect. 4.2 of the companion paper).

Discharge initialisation, a potential source of skill, is not considered. This has no effect on most of the analyses of the paper, since these are made in terms of run-off. Where discharge is analysed the effect of discharge initialisation is, due to the limited residence time of water in the rivers, restricted to the first lead month of the hindcasts (see Yuan, 2016 TS6).

## 3 Explanations of skill in hydrological variables

### 3.1 Skill in the meteorological forcing after bias correction

In this sub-section, the skill of the meteorological forcing will be analysed. Attention will be limited to the three input variables of VIC that have the largest effect on run-off and evapotranspiration, namely precipitation, 2 m temperature and incoming short-wave radiation. The WFDEI data are used as a reference. Here the data after bias correction are considered. In Appendix B we will discuss the skill of the raw S4 data, which is the meteorological forcing before bias correction. Differences in skill between the bias-corrected and the uncorrected data are negligible for temperature and short-wave radiation and small for precipitation.

Figure 1 shows results of the skill analysis of the precipitation forcing. Figure 1a provides an example of the skill for a single target and lead month (January as lead month 0). A summary of the skill in the precipitation hindcasts is given in Fig. 1b, which plots the fraction of all cells within the domain with statistically significant $R$ values, so Fig. 1a condenses into a single point in Fig. 1b. During the entire year, there is considerable skill for lead month 0 (on average in 61 % of the domain), but skill declines very rapidly to 6 % for lead months 1 and 2, just 1 % more than the percentage of cells in the case of no true skill at all. Hence, from lead month 1 on, skill is almost negligible. Regarding lead month 0, there is more skill in January, February and March than during the other months. For the lead month 0, hotspots of consistent skill, i.e. with a duration of significant skill of at least 3 target months, are situated on the Iberian Peninsula from November to March, in western Norway from January to April, in Greece and western Turkey from December to February, and in Scotland from December to March. All these occurrences of consistent skill are restricted to the winter half of the year

and mostly to coastal regions (see Fig. 1a), suggesting them to be linked to the initial state of the sea surface temperature.

Figure 2 shows important aspects of skill in the 2 m temperature hindcasts. One aspect is the possible contribution of a 30-year trend, which could be related to greenhouse warming, to the skill. Figure 2a and b provide summaries of the skill of the un-detrended and the detrended data, respectively, whereas Fig. 2c compares these two types of data. For lead month 0, the hindcasts have significant skill in the largest part of the domain (Fig. 2a and b), and detrending has a small effect (Fig. 2c). At longer lead times, the percentage of cells with significant skill quickly drops towards the theoretical no-skill limit (5 %), but there are a few exceptions, namely the following:

– For lead month 1, February and March temperatures are predicted with significant skill in a considerable part of the domain (44 % in February; 53 % in March). In both months the region with skill is more or less contiguous and comprises the Russian part of the domain, the Ukraine and the regions bordering the southern part of the Baltic Sea (Fig. 2d and e). In February the region of skill extends towards central Europe. In March it also comprises northern Fennoscandia. This skill hardly diminishes by detrending the data (Fig. 2b and c), suggesting that the skill is not related to climate change. Indeed, in February and March the observed trend (in the WFDEI data set) is insignificant across most of the domain (11 % of the domain in February and 18 % in March), and, more importantly here, it is insignificant in the regions with significant skill in the temperature hindcasts (Fig. 2g demonstrates this for March). We conclude that the temperature skill in February and March as lead month 1 must be due to initial conditions of the climate model (see also the discussion on Fig. 10).

– The 3 summer months (JJA) exhibit significant skill at all lead times in much more than 5 % of the domain (a range from 22 % to 56 % for all combinations of the 3 summer months and all lead months beyond lead month 0; see Fig. 2a). In this case the fraction of cells with significant skill is not a function of lead time, which is the type of behaviour that Yuan (2016) also found for the Yellow River basin. Since Fig. 2b and c demonstrate that the skill for JJA more or less vanishes when the temperature hindcasts and observations are detrended, we conclude that the skill for these months is due to trends in the data and is hence probably related to greenhouse warming. Another conclusion is that skill that hardly varies with lead time may be related to climate change.

It should be noted here that trends can only cause correlation between hindcasts and observations, and hence skill in the hindcasts, if they are present in both time series. A random time series of hindcasts is not correlated with a time series of observations with a trend and vice versa. Indeed, time series of both hindcasts and observations have a maximum in significant trends in summer, when trends form the prime source of skill according to our analyses. In the hindcasts and on average over all lead times beyond the first month, the summer months exhibit significant trends in almost the entire domain (95 %), versus 79 % of the domain in the other months of the year, on average. Similarly, observed trends are significant during the 3 summer months in 67 % of the domain, versus only 24 % of the domain in the other months of the year, on average. These percentages also show that significant trends occur in a larger part of the domain in the hindcasts than in the observations. So the observations, and not the hindcasts, are mostly limiting the occurrence of trend-related skill in the temperature hindcasts. This point is illustrated by the example of July as lead month 5 in Fig. 2f, h and i, but a similar illustration could have been provided for the other summer months and different lead months. Figure 2h shows that the trends of the hindcasts for July are significant across almost the entire domain (99 % of the domain). However, according to Fig. 2i only 69 % of the domain has a significant trends in the observed July temperatures. Indeed, the patterns of significance of Fig. 2f (skill in the temperature hindcasts) and Fig. 2i (significance of observed trends) agree to a large extent. The following points apply here: `CE7`

– April, May and September combine the behaviour of February and March, which have skill due to initial conditions of the climate model, with the skill of the summer months, which show skill related to trends (Fig. 2c).

– January has a considerable amount of significant skill but only for lead month 2 (42 % across the domain). This skill occurs in a piece of land reaching from England to Russia, which vaguely coincides with the region in which Kim et al. (2012) found skill in the S4 temperature hindcasts for the 3 winter months. However, as this skill is not found in adjacent lead and target months and is thus not consistent, we speculate that this skill is spurious.

Since short-wave incoming radiation is important for evapotranspiration, we finalise this sub-section with a short analysis of its predictability (Fig. 3). In terms of $R$, skill is considerable during the first lead month, with 58 % of the cells having significant skill, on average over the year. Months from March to September tend to have more skill than the other months of the year. Beyond lead month 0, skill settles around the no-skill line, except from April to July, but the fraction of cells with significant skill never exceeds 21 % (in May as lead month 1). Trends in the data hardly affect skill (Fig. 3b).

## 3.2 Sources of skill in run-off and discharge

In these sub-section analyses the effects of the meteorological forcing and the initial conditions on the predictability of

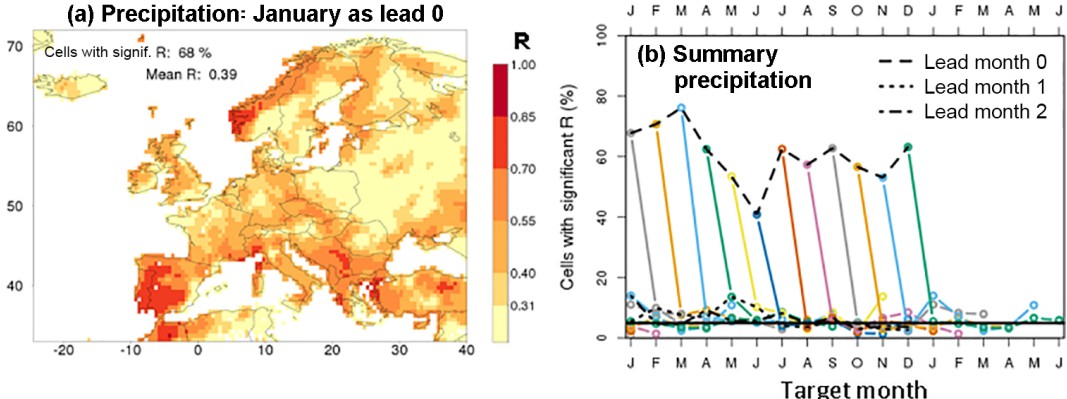

**Figure 1.** Skill TS7 CE5 of the precipitation hindcasts after bias correction. Figure 1a shows a map of the correlation coefficient between the observations and the median of the hindcasts ($R$), for target month January as lead month 0. The threshold of significant skill lies at 0.31, so cells with the lightest yellow colour have insignificant skill, and grid cells with other colours have significant skill. The legend provides the percentage of cells with significant values of $R$ and the domain-averaged value of $R$. Figure 1b depicts the percentage of cells with significant skill in terms of $R$ CE6 as a function of the target and lead month. Each coloured curve represents the hindcasts starting in a single month of the year and has a length of 7 (lead) months. For better visualisation the parts of the curves that end in the next year are shown twice, namely at the left-hand and the right-hand side of the graph. Black lines connect the results for identical lead times, which are specified in the legend (lead $m =$ lead month). The horizontal line gives the expected fraction of cells with significant skill due to chance in the case that the hindcasts have no skill at all (5 %).

run-off and discharge (discharge is only considered in Fig. 4) are isolated. We first address the question of how much of the skill in the run-off hindcasts is linked to trends. To examine this question, the pseudo-observations and the hindcasts of
5 run-off were detrended, and the skill was compared to that of the un-detrended data sets. We found that for lead month 2 and averaged over all months of the year, the fraction of cells with a significant $R$ decreased from 58.7 % to 57.4 % due to detrending, a difference of 1.3 %. This difference is much
smaller than the decrease for temperature (11.8 %). We conclude that trends contribute very little to skill in run-off. All analyses of this sub-section hereafter pertain to un-detrended data.

### 3.2.1 The relative importance of initial hydrological conditions

Figure 4 compares the InitSH with the FullSH in terms of the fraction of cells with a significant $R$ for run-off (Fig. 4a) and discharge (Fig. 4b). While the lumped results hardly differ between run-off and discharge (the companion paper
discusses small differences in skill between these two variables), systematic differences in skill between the FullSH and InitSH are revealed. For lead month 0, skill is higher in the FullSH than in the InitSH for all target months of the year, though the difference becomes very small when the fraction
of the domain with significant skill approaches 100 % and hence becomes unsuitable to discriminate between the two cases. Beyond lead month 1, the reverse occurs for most target months. Lead month 1 is transitional, with the order of skill depending on the time of the year. We produced figures

similar to Fig. 4, all shown in the Supplement, for the skill
evaluation of the following:

1. discharge with real, instead of pseudo-, observations, both for large basins (Fig. S1a in the Supplement) and small catchments (Fig. S1b) and for a subset of the large catchments with relatively little human impact (Fig. S2),

2. run-off in terms of the fraction of the domain with significant skill for the other metrics considered (RPSS, ROC above normal – AN – and ROC below normal (BN); Figs. S3–S5) and in terms of the domain-mean value of $R$ (Fig. S6).

In all of these cases, the reversal of skill around lead month 1 was found. So the reversal is a robust feature and is neither an artefact due to the type of observations nor due to human impacts on river flow, nor is it an artefact of the metric used in the verification procedure.

The explanation of the reversal deals with the ranking of the run-off in different years, since our metrics largely measure ranking. We will argue that while the InitSH forcing has a neutral effect on the ranking of the run-off forecasts and hence on their skill, FullSH forcing without skill has a
50 negative effect on the ranking of the run-off forecasts and hence on their skill. The InitSH forcing is, by construction, identical for all years. Using this forcing, inter-annual differences in forecasted run-off diminish with increasing lead time and approach zero when the effect of the initial condi-
55 tions vanishes. However, to a good approximation, rankings of forecasted run-off for different years remains the same as at $t = 0$. So the forcing has a neutral effect on the ranking

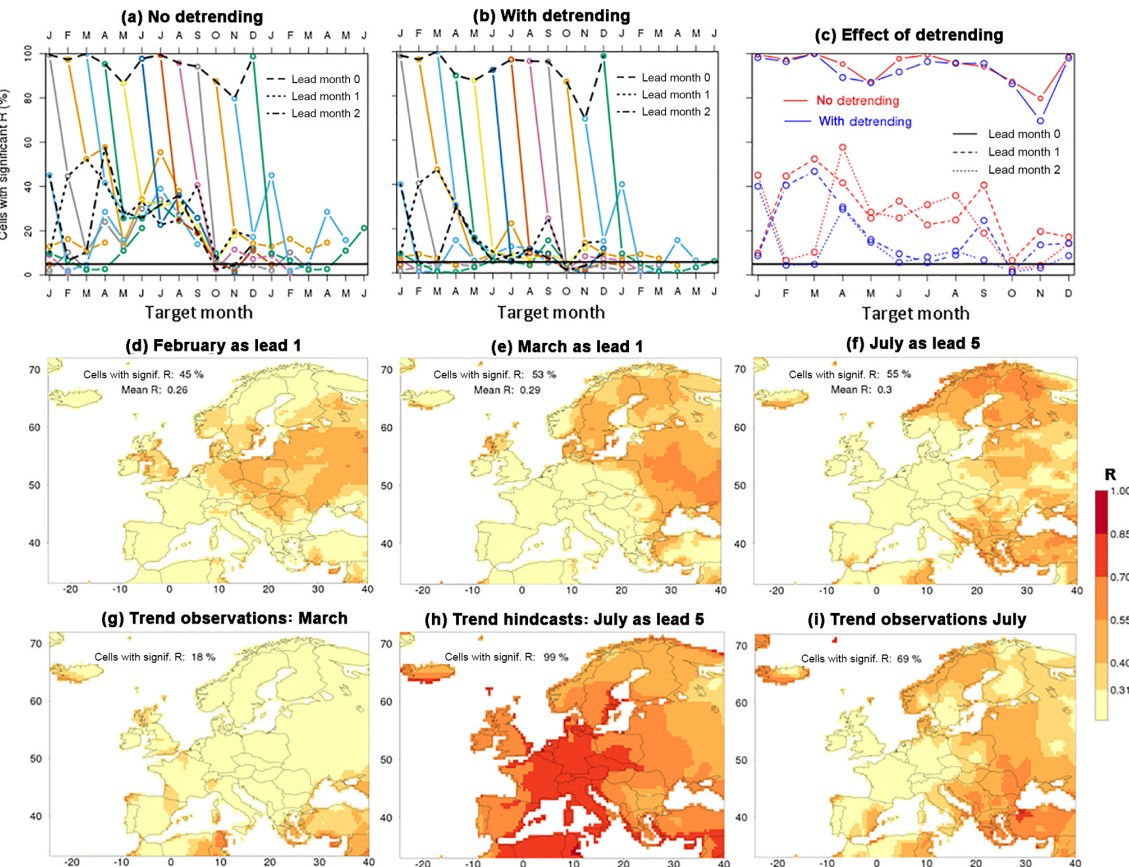

**Figure 2.** Skill of the 2 m temperature hindcasts after bias correction. **(a)** and **(b)** give the percentage of cells with significant values of $R$ for the un-detrended **(a)** and the detrended **(b)** temperature hindcasts (see Fig. 1b for further explanation). **(c)** compares annual cycles of skill of un-detrended and detrended data for the first 3 lead months. The three panels in the middle row show maps of $R$ for the un-detrended temperature hindcasts for target months February **(d)** and March **(e)** as lead month 1 and July as lead month 5 **(f)**. The bottom three panels depict the correlation coefficient of the trend (not the trend itself) of the observed monthly mean temperature, for March **(g)** and July **(i)**, and mean of the hindcasted temperature for July as lead month 5 **(h)**.

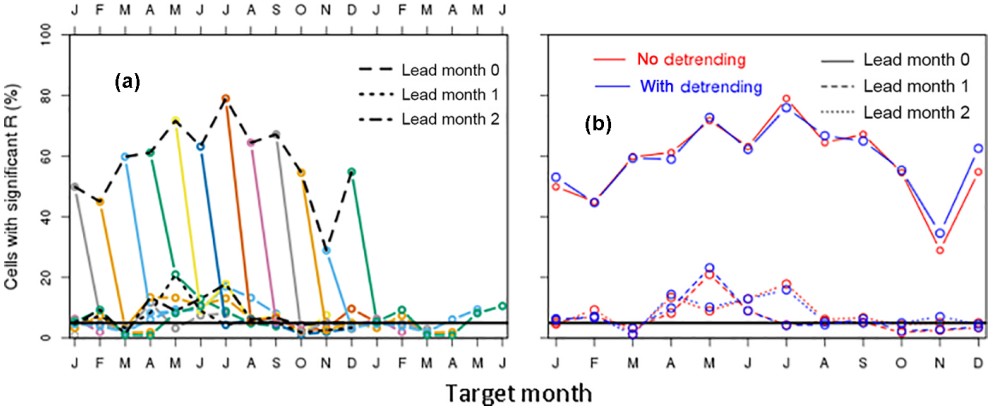

**Figure 3.** Skill of the incoming short-wave radiation hindcasts after bias correction. **(a)** gives the percentage of cells with significant values of $R$ (see Fig. 1b for further explanation). **(b)** compares annual cycles of skill of un-detrended and detrended data for the first 3 lead months (see **c** for further explanation).

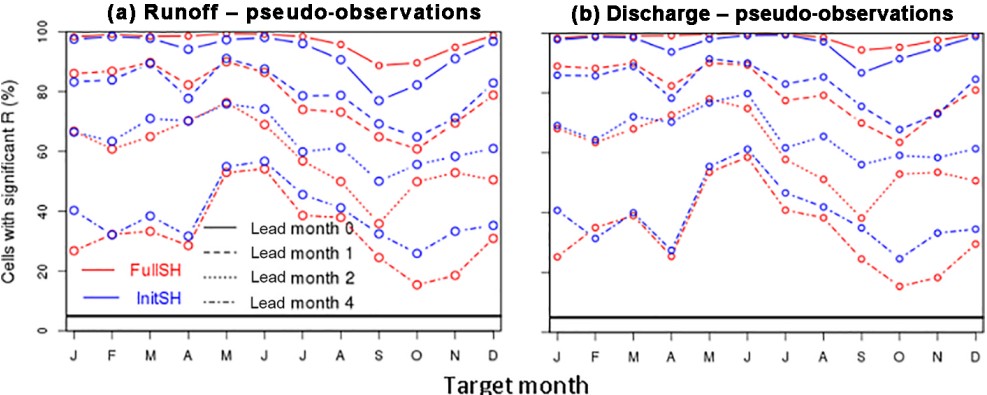

**Figure 4.** Comparison of the annual cycles of skill of the InitSH (blue) and the FullSH (red). The two panels show theoretical skill obtained with the pseudo-observations for run-off **(a)** and discharge **(b)** at four different lead times.

and hence on skill. Contrary to the InitSH, the FullSH forcing differs from year to year. This changes the ranking of different years of the run-off forecasts. If the FullSH forcings contain skill, these changes in ranking tend to bring, statistically, the forecasts towards the observations, so skill is added to the run-off forecasts. This is what happens at short lead times. At longer leads, the FullSH can be considered as having no skill. This tends to randomly shuffle the ranking of the run-off forecasts and hence diminishes their skill. Of course, the ranking of the (pseudo-)observations of different years also changes during the course of the forecasts, which generally has a negative effect on run-off skill unless forcing is perfect. This "observation argument" complicates the whole argument but it has no consequences for the argument above, since it affects the skill of the FullSH and the InitSH in the same way.

### 3.2.2 The relative contributions of soil moisture and snow initial conditions and of meteorological forcing

Figure 5 compares the skill in run-off of the specific hindcasts (except ESP) for 2 lead months (0 and 2). At both lead times and for all target months, initialisation of soil moisture is the dominant source of skill in Europe. Initialisation of snow and meteorological forcing are less important. This is true for all lead times (not shown here).

Meteorological forcing does not only have a relatively small contribution to the domain-averaged skill of Fig. 5 but also to regional skill. We searched for combinations of a region and target months where the MeteoSH produce consistently equal or more skill than the SMInitSH, but we did not find any combination where this was clearly the case. On average across the domain and for all target months, during the first lead month there is more skill due to the forcing (MeteoSH) than due to snow initial conditions (SnInitSH). For later lead months this order depends on the target month, mainly because skill due to snow initial conditions varies

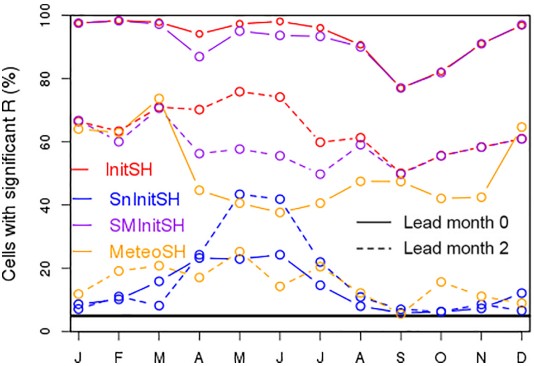

**Figure 5.** Comparison of the annual cycles of the skill in the run-off hindcasts of four specific hindcasts for lead months 0 and 2. Different colours correspond to different specific hindcasts, and different line types to different lead months.

strongly during the year. Although skill in run-off due to meteorological forcing (in the MeteoSH) is relatively small, it does exceed the skill in the forcing variable to which run-off is most sensitive, precipitation (compare Fig. 5 with Fig. 1). Whereas predictability of precipitation is almost limited to the first lead month, significant skill in run-off due to forcing is more widespread for lead months 1 and 2 (on average over the year, in 23 % and 15 % of the domain, respectively). We explain the enhanced skill in run-off mainly by an indirect effect. Skill in the precipitation forcing of the first lead month leads to skill in the states of soil moisture and snow at the end of that month. These model states then serve as the source of skill during the next lead months, when the precipitation forcing has no skill at all. In addition to this indirect effect of precipitation, the skill in the hindcasts of temperature (Fig. 2) contributes to skill in run-off in the MeteoSH.

From April to July, a considerable part of Europe has significant skill derived from snow initialisation, provided that initialisation does not occur earlier than in February, probably because in all parts of Europe with significant snow-

fall, this process does not stop before 1 February. Skill due to snow initialisation reaches a maximum in May and June, resulting in a maximum in skill in the InitSH hindcasts for these months and for most lead times. When snow contributes considerably to predictability (from April to July), the skill in the InitSH exceeds the skill in the SMInitSH. Because for target months from August to March snow contributes little to predictability, the percentages of cells with significant skill in InitSHs and SMInitSHs are almost identical for these months. The rapid rise in skill due to snow initialisation at the transition from April to May explains a remarkable feature that we noticed in the companion paper, namely an increase in run-off skill with lead time at this time of year. Another noticeable feature is that the skill due to snow initialisation for lead month 2 exceeds skill due to snow initialisation for lead month 0. This occurs for target months from May to August and will be explained in the text corresponding to Fig. 8.

Figures similar to Fig. 5, but for all metrics of the present study, are included in the Supplement (Fig. S7). The graphs for the ROC areas for the AN and BN `CE8` terciles are qualitatively similar to the graph for $R$. This also holds for the RPSS though fractions of the domain with significant RPSS are almost always lower than for the other metrics, probably because the RPSS is a summary metric for all three terciles including the middle one, which generally has much lower ROC areas than the other two terciles.

Figure 6 compares skill maps for the three specific hindcasts that isolate skill due to initial conditions (InitSH, SMInitSH and SnInitSH). It illustrates that skill due to snow and soil moisture initialisation is not only more or less additive at the scale of the entire domain (Fig. 5) but also at the regional scale. The patterns of skill due to soil moisture initialisation, e.g. in Africa, on the Iberian Peninsula and in western France (Fig. 5a), are also found in the map of skill due to both components of initialisation (Fig. 5c). Small regions with considerable skill due to snow initialisation (Fig. 5b), like those near Stockholm, in southeastern Czechia and southeastern Austria also stick out as foci of skill on the map of skill due to both soil moisture and snow initialisation (Fig. 5c). Where both soil moisture and snow initialisation cause moderate skill, e.g. in southern Finland, the combined specific hindcast exhibits more significant skill.

Figure 7 zooms in on the specific hindcast that isolates skill due to snow initialisation (SnInitSH), giving the example of a time series of skill as a function of lead time, after initialisation on 1 March. One observation is that skill does not gradually decrease with time but has a maximum during the snowmelt season. We like to note that locally skill is hardly generated during the part of the melt season when a snow pack covers the surface in each year. The reason is that in VIC the rate of snowmelt is almost insensitive to snow pack thickness (Sun et al., 1999). Hence, as long as the surface is covered by snow in each year, inter-annual vari-

ation in snowmelt is absent or negligible. Skill is only generated towards the end of the melt season, when snowmelt differs from year to year, because snow stops being available for melt at different dates due to different initial amounts of snow. So, the initial snow conditions cause skill because of inter-annual variation in the duration of the period that it takes to melt the snow present at the time of initialisation and not because of inter-annual variation in the melt rate. Of course, the timing of the end of the melt season differs regionally and with elevation, which largely explains the patterns of skill visible in the maps of Fig. 7. A good example is Scandinavia, where the earliest skill (in April; lead month 1) occurs at low elevations near the coasts of southern Norway and Sweden, at the end of the local snow season. The latest skill (in July; lead month 4) occurs in the Norwegian mountains, again at the end of the local snow season (we ascribe the skill in southeastern Sweden in July and August to chance). It is also relevant to note that the skill patterns in the maps of Fig. 7 are influenced by the fact that VIC has higher vertical resolution than its horizontal resolution may suggest, calculated `CE9` by performing simulations in multiple elevation bands within each grid cell, accounting for sub-grid variations in topography. Therefore, sub-grid topography leads to spreading of the snow skill signal of individual cells over longer periods of time.

To finish the analysis of the SnInitSH, Fig. 8 analyses a noticeable feature. In SnInitSHs, hindcasts for May have less skill when the hindcasts are initialised on 1 May (Fig. 8a) compared to initialisation during preceding months (February, March or April; Fig. 8b is for initialisation on 1 April). Similar results are found for June and July as target months. This result is noteworthy, because in hindcasts with initialisation on 1 May, there is, due to the use of pseudo-observations for verification, perfect knowledge about snow conditions on that date. With initialisation on 1 April, snow conditions on 1 May differ from those of the pseudo-observations, which by itself must lead to less skill in May run-off. The simple explanation is that on 1 April more grid cells have a snow cover than a month later on 1 May, but then the question arises of why those grid cells that lose their snow cover in April still exhibit significant skill in run-off during the month of May. The answer lies in an indirect effect. Inter-annual variations in the amount of snow at 1 April lead to predictable inter-annual variations in soil moisture on 1 May (Fig. 8c), when the snow cover has melted, which then acts by itself as an additional source of skill in run-off in May.

To finalise this section, the specific hindcasts were exploited to attribute the hotspots of significant skill in run-off for lead month 2, listed in the companion paper, to the different potential sources of skill. This was done for each of the hotspots by inspection of the maps of skill (like those of Fig. 6, for example) for three specific hindcasts that isolate the different sources of skill (SMInitSH, SnInitSH and MeteoSH). If the hotspot was present in, for example, SMInitSH, soil moisture initialisation is one of the sources

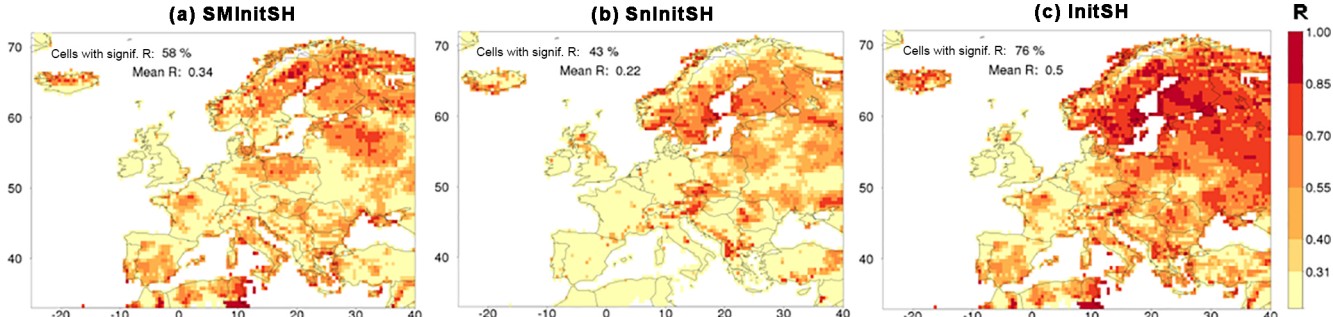

**Figure 6.** Example that compares the skill in run-off of three specific hindcasts (SMInitSH – **a**, SnInitSH – **b** and InitSH – **c**), for target month May as lead month 2. For more explanation, see Fig. 1a. White, terrestrial cells correspond to cells where observations or hindcasts consist for more than one-third of zeros or one-sixth of ties.

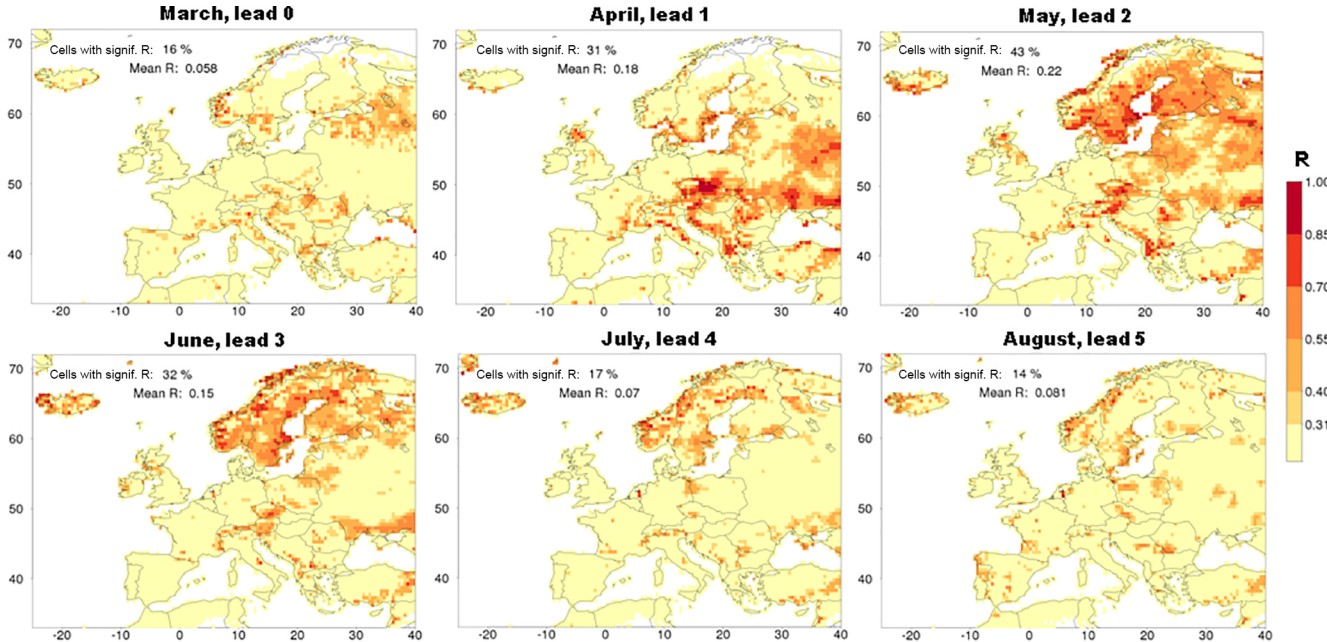

**Figure 7.** TS8 Example showing the variation of skill in run-off as a function of lead time in the SnInitSH, for initialisation on 1 March. For more explanation, see Figs. 1a and 6.

of skill. Results are summarised in Table 1. Almost all of the significant skill in the hotspot regions is due to the initial conditions of soil moisture. Exceptions are formed by the target months from April to July, when skill is caused by a mix of the initial conditions of snow and soil moisture in regions with significant snowmelt. In these cases the relative contributions of the two sources varies in time and space, but soil moisture is more important than snow, except in Fennoscandia, where snow dominates in June, and in July both sources are of about equal importance. Meteorological forcing contributed significantly to this in none of the hotspots of skill.

### 3.3 Skill and source of skill in evapotranspiration

This section analyses skill in the hindcasts of evapotranspiration, because hindcasts of evapotranspiration are use-

ful in themselves and evapotranspiration affects run-off (see Sect. 1), and in order to demonstrate the rich possibilities of the pseudo-observations, the specific hindcasts and the detrending to unravel the various sources of skill. In VIC, evapotranspiration is computed with the Penman–Monteith method (see Shuttleworth, 1993).

Figure 9a summarises skill in evapotranspiration in the FullSH. Levels of predictability are higher than for precipitation (Fig. 1), similar to those for temperature (Fig. 2) and lower than those for run-off (Fig. 4a). Figure 9b isolates the diverse contributions to skill for lead months 0 and 2 by showing the skill for the FullSH and three specific hindcasts. Averaged over the year, meteorological forcing (MeteoSH) contributes more to predictability in evapotranspiration than the initial conditions, among which soil moisture

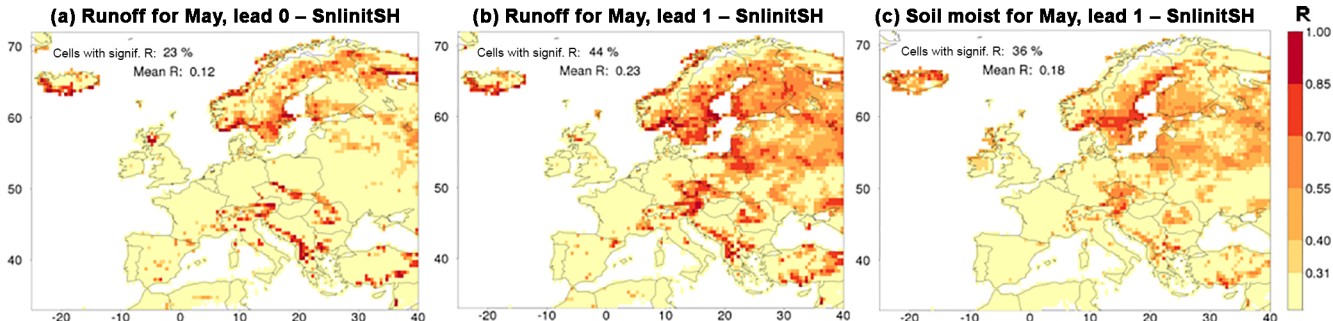

**Figure 8.** Example illustrating that skill in run-off for a target month may increase with lead time, namely for run-off in May as target months 0 (**a**) and 1 (**b**) in the SnInitSH. Skill in soil moisture in the SnInitSH, for May as lead month 1, is shown (**c**), because it provides part of the explanation for the mechanism causing the increase in skill with lead time. For more explanation, see Figs. 1a and 6.

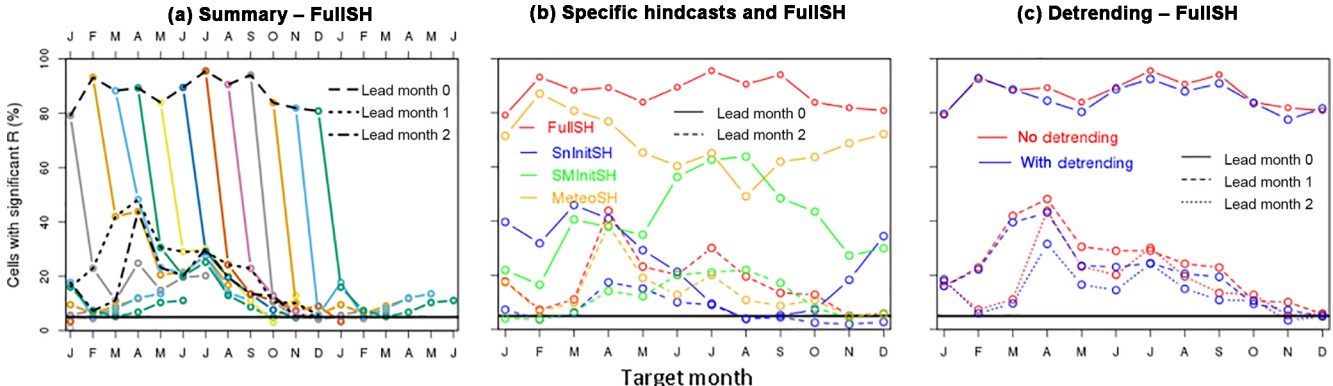

**Figure 9.** Summary plots of the skill of the hindcasts of evapotranspiration. (**a**) summarises the FullSH (for more explanation, see Fig. 1b), (**b**) depicts the annual cycles of skill for the FullSH and three specific hindcasts (SnInitSH, SMInitSH and MeteoSH), for lead months 0 and 2, and (**c**) compares the annual cycles of skill of the un-detrended and the detrended FullSH for the first 3 lead months.

**Table 1.** Sources of skill for hotspot regions and periods of skill. SM is soil moisture.

| Region | Period | Source of skill |
|---|---|---|
| Fennoscandia | Jan–Mar | SM |
| | Apr–Jul | SM and snow |
| | Aug–Oct | SM |
| Poland and northern Germany | Oct–Mar | SM |
| | Apr–May | SM and snow |
| Western France | Dec–May | SM |
| Romania and Bulgaria | Oct–Mar | SM |
| | Apr–May | SM and snow |
| Southern Mediterranean | Jun–Aug | SM |

(SMInitSH) causes more skill than snow (SnInitSH). Hence, comparing skill in run-off with skill in evapotranspiration, the most important source of skill shifts from the initial conditions of soil moisture to meteorological forcing.

In the FullSH (Fig. 9b) and focusing on lead month 2, there is hardly any skill in the evaporation hindcasts from

November to March (9 % of the domain, on average over these months), with the exception of January (18 %), when the region of skill (Germany and Benelux) is part of a larger region of skill in the temperature hindcasts for the same target and lead month. We blame the winter minimum of skill in evapotranspiration to the low levels of evapotranspiration and the low levels of skill in the temperature forecasts for the same period. The next month (April) exhibits the highest level of skill of all months (44 % of the domain), which is mainly due to meteorological forcing and has smaller contributions by the initial conditions of soil moisture and snow. From May to September there is some significant skill (23 % of the domain, on average over these months). Whereas in May forcing is still the most important contributor to skill, initial conditions of soil moisture form the main contributor from June to October. We speculate that this shift in the order of importance between forcing and soil moisture is due to the amount of variability in soil moisture. In Europe in spring (April and May), soil moisture variations are relatively small and hence hardly contribute to variations in evapotranspiration. Later in the year (June to September), soil moisture is often available in limited amounts, so variations are larger

and hence contribute more to variations in evapotranspiration. Snow initial conditions contribute to skill only during the snowmelt season from April to July.

The contribution of trends to predictability of evapotranspiration is summarised in Fig. 9c for lead months 0, 1 and 2. For lead month 2 and averaged over all target months of the year, detrending leads to a decrease in the fraction of cells with a significant $R$, from 17.6 % to 13.8 %, a difference of 3.8 %. The contribution of trends to skill in evapotranspiration is less than its contribution to skill in temperature (a difference of 11.8 %) but larger than its contribution to skill in run-off (a difference of 1.3 %). Trends contribute to skill in evapotranspiration during the part of the year when they also contribute to skill in atmospheric temperature (Fig. 2c), namely from April to September and in November (for lead month 0). However, whereas during the 3 summer months the skill in the temperature hindcasts is almost exclusively linked to climate change, a considerable part of the domain still exhibits skill in evapotranspiration after detrending.

To provide a deeper understanding of the skill in evapotranspiration, the skill in April and July is analysed in some detail. Figure 10 deals with April as lead month 2, showing the skill in evapotranspiration from the FullSH in Fig. 10a and from the MeteoSH in Fig. 10b. Regions of skill, mainly a piece of land from southern Fennoscandia to the Black Sea, are the same in the FullSH and in the MeteoSH, though skill is somewhat degraded in the MeteoSH. This indicates that meteorological forcing causes most, though not all, of the skill. Indeed, Figs. 2e (March) and 10c (April) show that the temperature forecasts for these two months after initialisation on 1 February contain skill in the mentioned region. We conclude that much of the skill in evapotranspiration is due to skill in the temperature hindcasts. The remaining part of the skill is due to initial hydrological conditions. While Fig. 9b shows this for the entire domain, we also found limited amounts of skill in the SnInitSH and the SMInitSH for April in the stroke of land from southern Fennoscandia to the Black Sea (not shown here). This means that in that region, initial conditions of the hydrological model on 1 February provide some skill to the hindcasts of evapotranspiration for April. We like to note that this could be consistent with the conclusion in Sect. 3.1 that the skill in the temperature hindcasts of February and March in this same region are due to the initial conditions of the climate model. These initial conditions could be, for example, sea surface temperatures or also CE10 the local state of snow and/or soil conditions. In the latter case, the two types of predictability in the mentioned regions would have the same or a similar source. Initial conditions of snow and/or soil conditions in S4 would lead to skill in the temperature hindcasts of S4, while initial conditions of snow and soil moisture in VIC lead to skill in the evapotranspiration hindcasts of VIC.

During the summer months and for all lead times, skill in evapotranspiration occurs in two regions, namely the southern part of the Mediterranean and western and northern Nor-

way. Figure 11 shows target month July for lead month 5 as an example. Whereas Fig. 11a is for the FullSH, Fig. 11b–d depict the maps for three specific hindcasts (SnInitSH, SMInitSH and MeteoSH), and Fig. 11e shows skill for the FullSH after detrending. Since the SnInitSH and the MeteoSHs exhibit hardly any skill, while the SMInitSH has considerable skill in the Mediterranean (Fig. 11b–d), it can be concluded that the skill in this region is due to soil moisture initial conditions. So, in this particular case, knowledge of soil moisture conditions on 1 February still yields skill in evapotranspiration in July. This skill in the Mediterranean is not affected by detrending (compare Fig. 11a and e), so it does not have a climate-change component.

The skill in Norway has a more complicated origin. The three specific hindcasts show that it is due to a mix of initial snow conditions (Fig. 11c) and meteorological forcing (Fig. 11d). The effect of the initial snow conditions (on 1 February) can be understood with the help of the analysis of run-off skill in the SnInitSH (Fig. 7), which led to the conclusion that run-off skill caused by snow initialisation occurs at the end of the melt season, which is July in much of Norway. Therefore, in this country and in July the timing of the disappearance of snow cover varies from year to year. This then has a considerable effect on evapotranspiration, since bare soil has, compared to snow, higher surface temperatures and hence more evapotranspiration in summer. The contribution to skill by forcing (Fig. 11d) fades with, but is not removed by, detrending (not shown here), so it has a part that is related to climate change and a part that is unrelated to climate change. The climate-change-related skill due to forcing resides in the temperature hindcasts, which have significant skill in this region at all lead times (Fig. 2f). The non-climate-change-related skill in the MeteoSH for July is likely an indirect effect of the skill in the forcing (especially precipitation) during the first lead month (February). This leads to skill in snow water that is equivalent towards the end of February, which fades but has not disappeared completely on 1 July (Fig. 11f) and then causes skill in evapotranspiration at the end of the melt season.

## 4 Discussion

### 4.1 Comparison of skill with previous studies

A remarkable result of our work is the reduction of the skill in run-off beyond lead month 1, when annually varying S4 forcing is used (FullSH) instead of meteorological forcing that is identical for all years (InitSH; see Fig. 4). This result is counter-intuitive but is, as we discussed, a logical consequence of forcing with inter-annual variation that has no or insufficient skill, such as the S4 forcing. Other studies compared the FullSH (also called climate-model-based hindcasts) with ESP hindcasts, which are slightly different from our InitSH (see Sect. 4.3) but like the InitSH have uninforma-

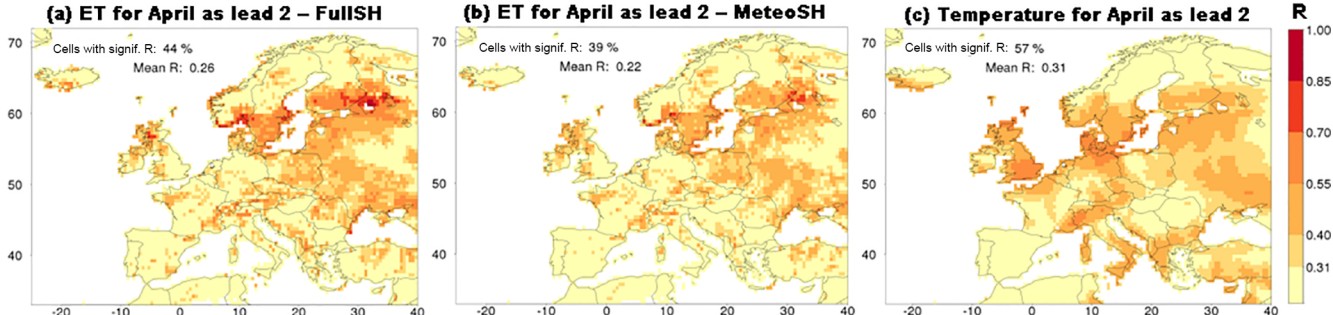

**Figure 10.** Explanation of the skill in the hindcasts of evapotranspiration, for target month April as lead month 2. The panels map the skill in evapotranspiration of the FullSH (**a**), of the MeteoSH (**b**) and of the hindcasts of temperature (**c**). For more explanation, see Fig. 1a.

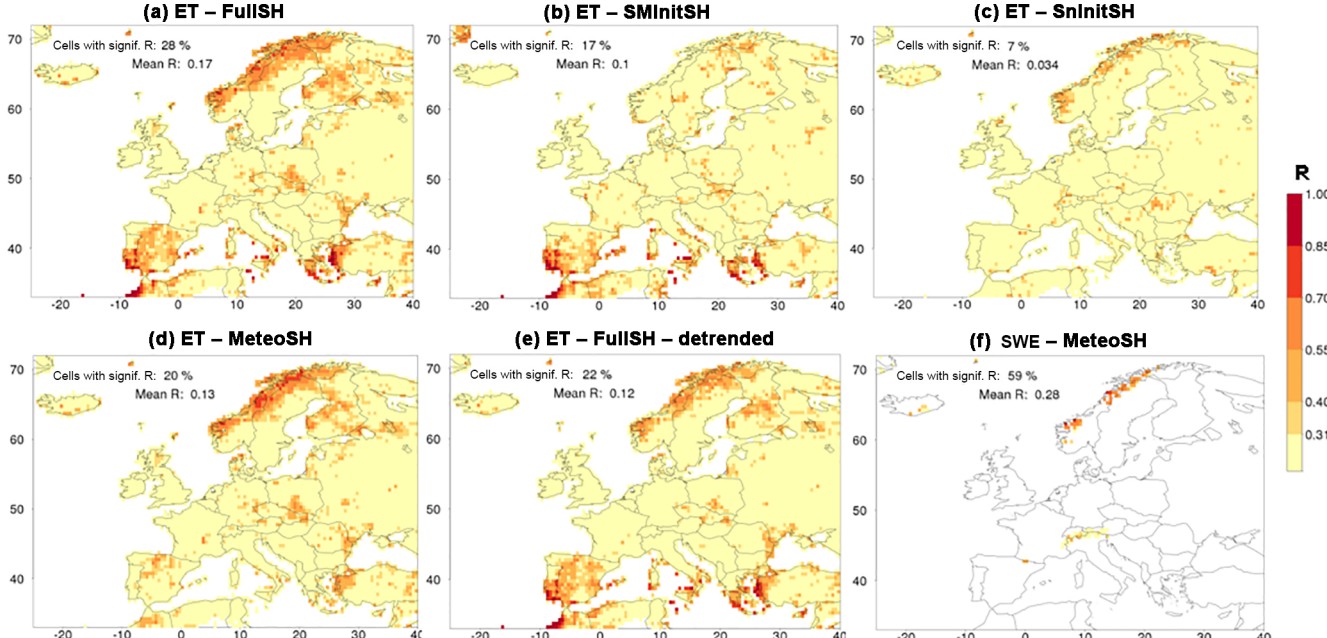

**Figure 11.** Explanation of the skill in the hindcasts of evapotranspiration (ET) for July by taking lead month 5 as an example. The panels map the skill in evapotranspiration of the FullSH (**a**), SMInitSH (**b**), SnInitSH (**c**), MeteoSH (**d**) and the FullSH after detrending (**e**). (**f**) depicts skill of the hindcasts of snow-water equivalent (SWE) in the MeteoSH. For more explanation, see Fig. 1a. Note that statistics in the legends of the panels refer only to that part of the domain for which $R$ was computed, which consists of all coloured cells.

tive meteorological forcing for each year. Some of these studies (e.g. Singla et al., 2012; Mackay et al., 2015) found little overall difference in skill between the FullSH and ESP hindcasts. However, in a study of Canadian catchments, Bazile et al. (2017) broadly confirm our finding that beyond the first lead month, ESP is superior to the FullSH, while the reverse holds for the first lead month. Arnal et al. (2018) TS9 compared FullSHs with ESP hindcasts and found that, in Europe, ESP has more discrimination skill ("potential usefulness") than FullSHs, although there are exceptions both spatially and seasonally. These authors, however, do not mention any trend with lead time in the difference between FullSHs with ESP. In contrast with our results, skill is enhanced when using meteorological hindcasts and also at longer leads, like

CE11 in the studies of Yuan et al. (2013), Thober et al. (2015), Yuan (2016) and Meißner et al. (2017). This contrast might be explained by more skill in the meteorological hindcasts of the mentioned studies than in the present study, which could be due to the type of meteorological hindcasts (only Meißner et al., 2017, used S4) or the investigated region (in the mentioned studies of the US, Europe, China and Germany, respectively). Europe is a region with relatively little skill in meteorological hindcasts (Kim et al., 2012; Scaife et al., 2014; Baehr et al., 2015). Effects of regional differences in the skill of the forcing on the relative skill of FullSHs and ESP are mentioned by Wood et al. (2005), who reported that FullSHs for the western United States have practically no skill improvement over the ESP, except for some regions

and seasons with predictability of the forcing originating in El Niño–Southern Oscillation CE12 teleconnections.

The specific hindcasts of this study show that in Europe initial conditions of soil moisture are the largest source of skill in the seasonal run-off forecasts produced with WUSHP. Contributions to skill by the initial conditions of snow and by the meteorological forcing are mostly much smaller. To our knowledge, two other studies analysed sources of skill of hydrological seasonal forecasts for Europe with dynamical systems similar to those of the present study, namely Bierkens and Van Beek (2009) and Singla et al. (2012). Comparing our results with those of Bierkens and van Beek (2009), both studies agree that initial conditions form the dominant source of skill. However, compared to the present study, Bierkens and van Beek (2009) find a larger contribution to skill by the meteorological forcing, at least in summer. This difference might be due to the quality of the forcing. Bierkens and van Beek (2009) developed an analogue events method to select, on the basis of annual sea surface temperature (SST) CE13 anomalies in the North Atlantic, annual ERA40 meteorological forcings, which they used as forcing for their hydrological model. One might speculate that in Europe their semi-statistical forcing is more skilful than the S4 forcing used in WUSHP. This suggests that there is room for improvement of climate-model seasonal forecasts, so if and when this improvement is realised, the relative contribution of the meteorological forcing to skill in hydrological variables would increase. As for the second study of the sources of skill, conclusions of Singla et al. (2012) are not directly comparable with those of the present study, as they used ESP and reverse ESP (see Sect. 4.3).

## 4.2 Understanding the skill due to initial soil moisture

The dominance of soil moisture initial conditions in terms of domain-lumped skill also extends to the hotspot regions and periods of skill (Table 1). The understanding of the skill linked to soil moisture can be deepened by another level, as in Shukla and Lettenmaier (2011). The underlying idea is that this type of skill increases with the inter-annual variability of soil moisture at the date of initialisation and that this skill is gradually eliminated during the course of the hindcasts by inter-annual variability in processes like rainfall and snowmelt. The question is the following: to what extent are the hotspots of skill (see Table 1) linked to soil moisture initialisation due to the cause of the skill, and to what extent they are due to a lack of inter-annual variability in the processes that eliminate the skill? Figure 12 helps answer this question for the skill found in the run-off hindcasts of August as lead month 2 with a simple method of analysis. Figure 12a shows the standard deviation of total modelled soil moisture ($\sigma_{SM}$) on the day of initialisation (1 June), taken from the reference simulation. Figure 12b depicts the standard deviation of total rainfall ($\sigma_{RF}$) during the course of the hindcast (June–August), taken from the WFDEI data set, which

is the investigated skill-eliminating factor. These two quantities were combined into an estimate of the skill ($S_{est}$):

$$S_{est} = \exp\left( -\frac{\sigma_{RF}^2}{\sigma_{SM}^2} \right). \tag{1}$$

This estimate (Fig. 12c) needs to be compared with the skill of the hindcasts, mapped in Fig. 12d in terms of $R$. The two maps are not expected to be exactly equal, not only because of the simplicity of the estimation method but also because $S_{est}$ is not a correlation coefficient. However, in the limits, $S_{est}$ has the desired properties. It is equal to zero for the cases of constant initial amounts of soil moisture or infinite variability in rainfall. It is equal to 1 for the cases of infinite variability in soil moisture or constant rainfall. The correlation coefficient between the patterns in Fig. 12c and d is highly significant (0.67), and the hotspot regions of skill are the same in both panels, namely the northern part of Fennoscandia and the southern part of the Mediterranean. So, in the case of August as lead month 2 the estimation method is reasonably successful in computing the pattern of skill in the hindcasts with the simple means of the WFDEI data set and model calculations from the reference simulation. The merit of the estimation method is the deeper understanding of the cause of the skill in the two hotspot regions. Northern Fennoscandia is a hotspot, because the amount of inter-annual variability in initial soil moisture is larger than elsewhere (Fig. 12a). The southern part of the Mediterranean is a hotspot, because the amount of inter-annual variability in rainfall is lower than elsewhere (Fig. 12b).

This simple method of analysis helped to bring the understanding of the skill in northern Fennoscandia and the southern Mediterranean to a deeper level, but it was less successful for the other hotspots. A more thorough analysis along these lines and a deeper understanding of skill in the hindcasts is left for future work.

## 4.3 Relation of the present specific hindcasts with conventional ESP

The specific hindcasts of this study are related to the well-known ESP (e.g. Wood and Lettenmaier, 2008; Shukla and Lettenmaier, 2011; Singla et al., 2012; Van Dijk et al., 2013; Harrigan et al., 2018 TS10). ESP is not only used as an experimental tool in science but is also widely used to produce forecasts in operational mode (Day, 1985). ESP used for scientific purposes can be subdivided into proper ESP (called ESP from now on) and reverse ESP.

ESP (hindcasts) is similar to the InitSH of this study. In both types of hindcasts the initial conditions vary from year to year and are quasi-perfect, i.e. they are taken from a simulation like our reference simulation, while the meteorological forcing is uninformative, e.g. by being the same for all years (in the InitSH and, for example, in the ESP of Shukla and Lettenmaier, 2011) or by varying randomly from year to year

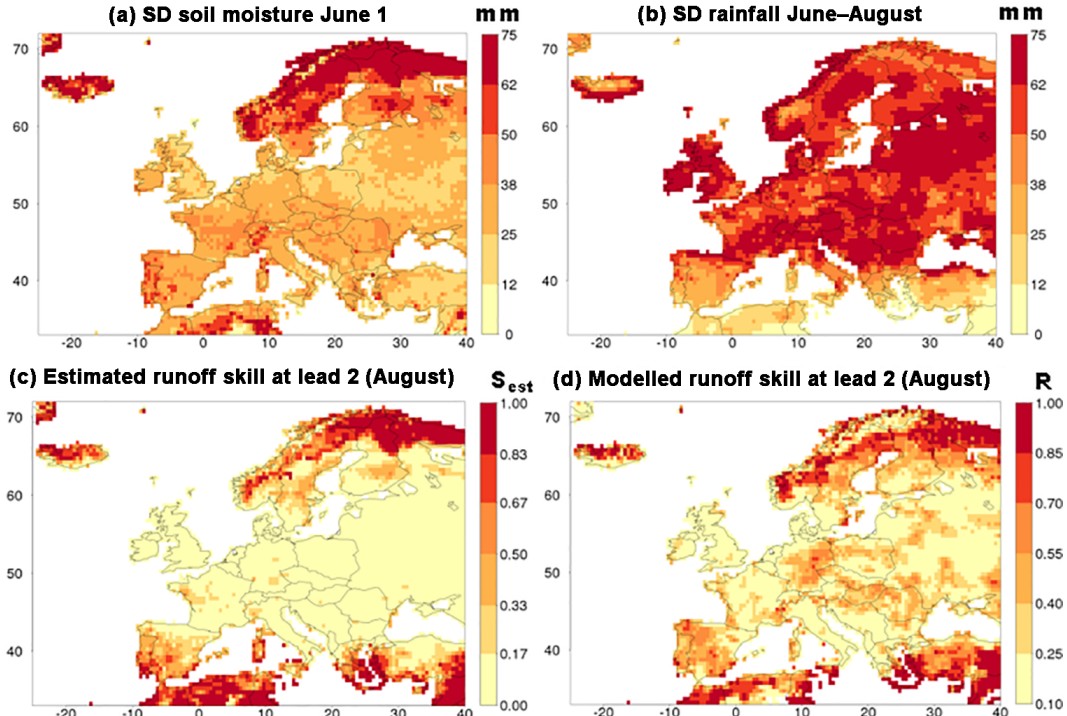

**Figure 12.** Illustration of a simple method that partly explains skill in run-off due to initial soil moisture, exemplified for target month August as lead month 2. **(a)** is a map of the standard deviation in soil moisture at the date of initialisation (1 June). Similarly, **(b)** maps the standard deviation of observed rainfall during the course of the hindcasts (June–August). These two standard deviations are combined into an estimate of the skill (Eq. 1) in **(c)**, which is compared with the skill of the FullSH **(d)**. Note that the colour scales of **(c)** and **(d)** differ from each other and differ from scales of other figures (e.g. Fig. 1a).

(e.g. in the ESP of Singla et al., 2012). This eliminates skill due to the meteorological forcing, so skill can only be due to the initial conditions. However, while in ESP the forcing is selected from historic observations, it is selected from the S4 hindcasts in InitSHs in order to retain an inter-member variability and other statistical characteristics of the time series similar to that in the FullSH. An advantage of ESP is that its production is relatively cheap, because no climate-model forecasts are needed.

Similarly, reverse ESP (see Wood and Lettenmaier, 2008) resemble the MeteoSH of this study. In both types of hindcasts the meteorological forcing varies from year to year, while the initial conditions are identical for each year. This eliminates skill due to the initial conditions, so skill can only be due to the forcing. However, while in reverse ESP the forcing of each year is made up of the observations of that year, it is made up of the S4 hindcasts in the MeteoSH. Moreover, in reverse ESP, ensembles are built by using differing initial conditions, whereas they are built by using differing meteorological forcings in the MeteoSH.

In ESP and in the InitSH, if all skill due to the meteorological forcing is indeed removed, the remaining skill, which is due to the annually varying initial conditions, should logically be the same in both types of hindcasts, since the initial conditions are the same. To test this expectation we produced

ESP and compared its skill with that of the InitSH. Indeed, skill from these two types of hindcasts is almost identical, as demonstrated in the Supplement (Fig. S8). We conclude that skill produced with specific hindcasts with a forcing that does not vary from year to year is not sensitive to the choice of that forcing, perhaps with the exception of forcings that deviate strongly from being realistic. We like to note here that, in odd years, one of the ESP ensemble members is identical to the pseudo-observation used for verification. This is a concern, but we deemed this less important than the requirement of identical forcing for all years, which is crucial for the explanation of the skill reversal (Sect. 3.2.1).

This similarity of the InitSH and ESP is in sharp contrast with the skill resulting from reverse ESP and MeteoSHs, which are expected to be totally different. Keeping in mind that in both types of hindcasts, skill is caused only by skill of the meteorological forcing, this is the skill of the S4 hindcasts in the MeteoSH. The present study showed that in Europe there is a small contribution to skill in the run-off hindcasts by the forcing and that this contribution tends to decrease with time. This differs from reverse ESP, in which skill is small at the beginning and then increases with lead time to reach perfect skill at very long leads (see Wood and Lettenmaier, 2008), because the meteorological forcing is quasi-perfect (i.e. identical to the forcing in the reference simula-

tion), while the influence of the initial conditions, which are non-informative in reverse ESP, decreases with time.

## 4.4 Towards an operational system

We plan to launch an operational version of WUSHP. That version might include a post-processing procedure with the aims of removing biases in discharge and making the system more reliable. This could perhaps be done with statistical calibration (e.g. Gneiting et al., 2005; Schepen et al., 2014), a technique that, contrary to quantile mapping, considers information that is available from correlations between hindcasts and observations (see Wood and Schaake, 2008; Madadgar et al., 2014).

The superiority of the InitSH (and the ESP) with respect to the FullSH for hindcasts beyond the first 2 lead months raises the question of whether one should, in an operational version of WUSHP and for these lead months, issue forecasts like the InitSH (or ESP) and not forecasts like the FullSH. The logical answer is "yes", but such a strategy should then be reconsidered when the meteorological forcing is taken from a new, possibly improved version of the climate model or from another, possibly better type of climate model.

The applied methods of analysis are not suitable for giving quantitative advice on what would be the best investment for increasing the amount of skill of WUSHP. However, since initial soil moisture is the dominant source of predictability, a large gain of skill could possibly be made by assimilation of soil moisture observations into the modelled state of soil moisture (see e.g. Draper and Reichle, 2015). In addition, observations of a snow-water equivalent could be assimilated into the modelled state of snow (see e.g. Griessinger et al., 2016). Improving the calibration of VIC would be another obvious road towards improvement of the seasonal predictions discussed in this paper. This should lead to higher actual skill but not necessarily to more theoretical skill (see the discussion section of the companion paper).

## 5 Conclusions

The present paper explains skill in the hindcasts of WUSHP, a seasonal hydrological forecast system, applied to Europe. We first analysed the meteorological forcing, which consists of bias-corrected output from a climate model (S4), and found considerable skill in the precipitation forecasts of the first lead month but negligible skill for later lead times. Seasonal forecasts for temperature have more skill. Skill in summer temperatures was found to be related to climate change occurring in both the observations and the hindcasts, and it was found to be more or less independent of lead time. Skill in northeastern Europe in February and March is unrelated to climate change and must hence be due to initial conditions of the climate model.

Sources of skill in run-off were isolated with specific hindcasts, namely SMInitSHs (soil moisture initialisation), SnInitSHs (snow initialisation), InitSHs (a combination of soil moisture and snow initialisation) and MeteoSHs (meteorological forcing). These hindcasts revealed that, beyond the second lead month, hindcasts with forcing that is identical for all years but with "perfect" initial conditions (InitSH) produce, averaged across the model domain, more skill in run-off than the hindcasts forced with S4 output (FullSH). This occurs because inter-annual variability of the S4 forcing adds noise, while it has hardly any skill. The other specific hindcasts showed that in Europe initial conditions of soil moisture form the dominant source of skill in run-off. For target months from April to July, initial conditions of snow contribute significantly, with a domain-mean maximum in May and June. The timing of that maximum varies spatially and coincides with the end of the melt season, when snowmelt differs from year to year, because snow stops being available for melt at different dates. All regional and temporal hotspots of skill in run-off found in the companion paper are due to initial conditions of soil moisture, with smaller or larger contributions by the initial conditions of snow for target months from April to July in hotspot regions with snowfall in earlier months. We further showed that skill due to snow and soil moisture initialisation is more or less additive.

Some remarkable skill features are due to indirect effects, i.e. skill due to forcing or initial conditions of snow and/or soil moisture is, during the course of the model simulation, stored in the hydrological state (snow and/or soil moisture), which then by itself acts as a source of skill.

Predictability of evapotranspiration was analysed in some detail. Levels of predictability and the annual cycle of skill are similar to those for temperature. For most combinations of target and lead months, forcing forms the most important contributor to skill, but for lead month 2, initial conditions of soil moisture dominate from June to October.

*Data availability.* . TS11

## Appendix A: Reliability of the hindcasts

To complement the analysis of discrimination skill of WUSHP published in the companion paper, this appendix presents a short evaluation of the reliability of the system. Per definition, forecasts are considered "reliable" when the forecast probability is an accurate estimation of the relative frequency of the predicted outcome (Mason and Stephenson, 2008). We assessed the reliability of the discharge hindcasts of the FullSH by means of so-called reliability diagrams (see Mason and Stephenson, 2008), which we produced and evaluated as follows.

– For each grid cell and combination of a category (or tercile; AN, NN and BN), lead month and target month we proceeded as follows:

– Divide the 30 (number of years) observations into terciles and give them a binary number (1 if the event falls in the considered category and otherwise 0).

– Divide the 450 (number of years × number of ensemble members) forecasts into terciles.

– Determine, for each of the 30 years, the forecast probability of the event occurring (forecast falling in the considered tercile).

– Pair the binary observations with the forecast probabilities.

– Sort the paired data into eight bins stratified by the forecast probabilities of the event.

– Compute bin averages of the forecast probability and of the binary observations.

– Pool the results for two consecutive lead months and the 3 target months of the same season.

– The results were further processed as follows:

– They were aggregated for the entire domain and then plotted. Examples for the BN tercile and the spring months (MAM) as a target are shown in Fig. B1a–c, with the lead month number increasing from left to right. In each diagram a linear regression is applied to the data points, weighing individual points by the number of data pairs in the bins. Because tercile thresholds are set independently for observations and forecasts, the resulting line always goes through the climatological intersection (one-third in our case; see Weisheimer and Palmer, 2013[TS13]), and results are insensitive to biases. As in Weisheimer and Palmer (2013[TS14]) we use the slope of the line as a measure of reliability. A slope equal to 1 corresponds to perfect reliability, and a slope equal to 0 indicates no reliability at all.

– Reliability diagrams similar to those in Fig. A1a–c were produced for each terrestrial grid cell, and best-fit lines and their slopes were computed. The slopes were plotted in maps, of which examples for the BN tercile and the spring months (MAM) as a target are shown in Figs. A1d–f and A2d–f[TS15].

For the analysis it is helpful to first consider the value of the slope in two extreme cases. If pseudo-observations are used for verification and lead time approaches zero, all members of the hindcasts for a specific year approach the pseudo-observation of that year. Hence, all hindcasts fall in the same category as the observation, so the reliability diagram condenses to two points at the coordinates [0, 0] and [1, 1][TS16], which represent, respectively, two-thirds and one-third of all contributing data. In this case the hindcasts are utterly reliable and utterly sharp. The second case is when the hindcasts have no discrimination skill at all, i.e. forecast probabilities of an event are randomly paired with the outcome (whether the event occurs or not). In this case, the slope of the fitted line is equal to zero, so the hindcasts are not reliable at all, and sharpness is minimal, i.e. forecast probabilities tend to approach one-third for each of the terciles.

In Fig. A1 reliability is evaluated for the case of verification with pseudo-observations. For the first 2 lead months, the slope of the line in the diagram of the aggregated data (Fig. A1a) is 0.916. Hence, during these 2 lead months the system is not far from being perfectly reliable, and it is rather sharp with relative maxima in forecast probability in the lowest and the highest bin. Then, with progressing lead time, reliability is reduced, i.e. the slope of the aggregated data decreases to 0.767 (for lead months 2 and 3; Fig. A1b) and 0.469 (for lead months 4 and 5; Fig. A1c). Moreover, with increasing lead time, sharpness is reduced, with gradually more ensemble forecasts approaching the climatological forecast, i.e. a probability of one-third for each of the terciles.

The maps of Fig. A1d–f show the geographical distribution of the slope from the reliability diagrams. For the first 2 lead months most values of the slope for individual grid cells lie between 0.7 and 1.1 (Fig. A1d), and the domain-averaged slope is 0.910. At longer leads, the highest values are found in some regions with considerable amounts of discrimination skill, such as Poland and northern Germany, western France, and Romania and Bulgaria (see Table 1). Reliability also tends to increase towards the northeast of the continent. Domain mean values of the grid-level slope are generally somewhat lower than the slope of the aggregated data. This can, at least partly, be ascribed to more scatter of individual points around the best-fit line because of the much smaller sample size for individual grid cells.

Reliability for the AN tercile is almost equal to that for the BN tercile, while slopes are much closer to zero for the NN tercile (not shown here). Also, levels of reliability show little variation during the year, except for the autumn (SON), when slopes are smaller (not shown here). Finally, Fig. S9

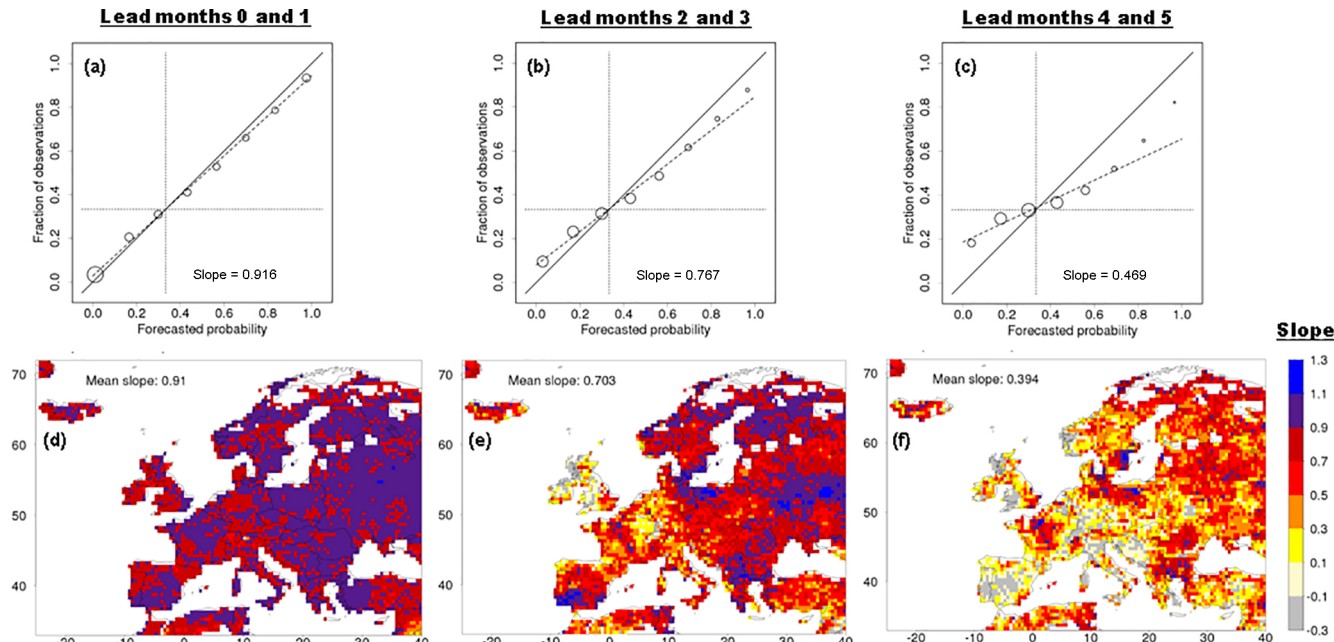

**Figure A1.** Reliability of the FullSH discharge hindcasts for the BN tercile in spring (target months MAM). Pseudo-observations were used for verification. Lead time increases from left to right TS12. **(a–c)** depict aggregated reliability diagrams for the full domain. The forecasted probabilities of BN discharge (horizontal axis) are collected in eight bins. The vertical co-ordinate is the relative frequency of BN discharge observations for all of the forecasts in a specific bin. The solid line is the 1 : 1 line. The dashed line shows the best fit to the eight data points, each weighted by the number of observations contributing to the bin ($N_{bin}$). The area of the symbols is proportional to $N_{bin}$. The dotted lines are the averages of the variables along the two axes (one-third). Similar reliability diagrams were made for all grid cells individually, and the slopes of the best-fit lines are plotted in **(d)**–**(f)**.

shows that for verification in CE14 real instead of pseudo-observations, slopes are closer to zero, so forecasts seem to be less reliable and more overconfident. Strikingly, discrimination skill and reliability have similar characteristics. Both decrease with increasing lead time, and differences between the AN and BN terciles are relatively small, while scores for the NN tercile are clearly inferior to those for the two outer terciles. Also, regional maxima in discrimination skill and reliability tend to coincide, and scores of discrimination skill and reliability are smallest in autumn.

## Appendix B: Skill in the meteorological forcing before bias correction

Section 3.1 contains an analysis of the skill of the meteorological forcing after bias correction. Because predictability of the meteorological forcing is an interesting topic by itself, we present here an analysis of the skill of the meteorological forcing before bias correction, i.e. of the raw S4 output, again limiting attention to the three variables considered in Sect. 3.1. Figure B1a summarises the skill of the raw precipitation hindcasts, which should be compared with the summary for the bias-corrected hindcasts of precipitation in Fig. 1b. Such a comparison is made for lead months 0, 1 and 2 in Fig. B1b. Similar comparisons are made for the 2 m tem-

perature and incoming short-wave radiation in Fig. B1c and d, respectively. At this level of summarising the differences in skill between the two types of data, differences are small for precipitation and negligible for temperature and short-wave radiation. Also, patterns of skill for all three variables, such as those shown in the maps of Figs. 1 and 2, are almost identical for the bias-corrected and raw data. The fact that differences are small is not surprising, because the bias corrections hardly change the ranking of the values, while the value of the correlation coefficient largely depends on the ranking of the hindcasts relative to the ranking of the observations. Results, in terms of differences in skill between raw and bias-corrected meteorological forcing, are essentially the same for the other metrics used (ROC area and RPSS).

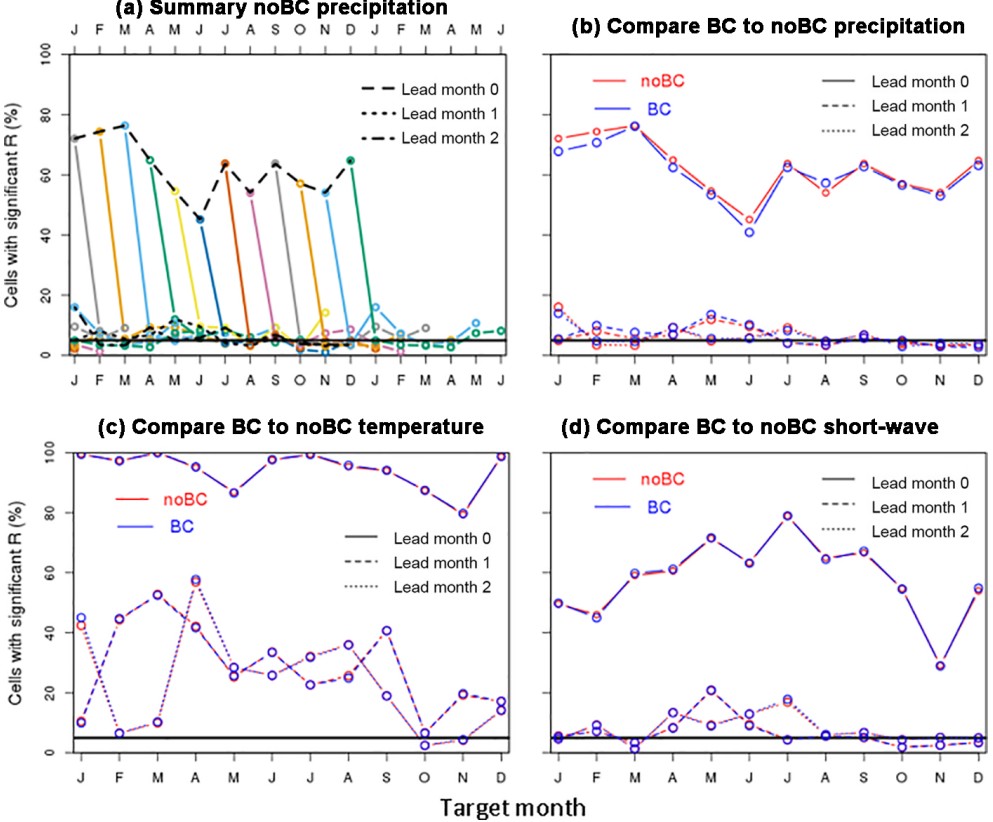

**Figure B1.** Skill, in terms of the percentage of cells with significant values of *R*, for three components of the raw S4 forcing. **(a)** shows precipitation skill as a function of target and lead month. The other three panels compare the skill of the raw S4 output (noBC), with its bias-corrected version (BC) as a function of the target month CE15 for the first 3 lead months. Precipitation is plotted in **(b)**, temperature in **(c)** and incoming short-wave radiation in **(d)**.

*Supplement.* The supplement related to this article is available online at: https://doi.org/10.5194/hess-23-1-2019-supplement.

*Author contributions.* .TS17

*Competing interests.* The authors declare that they have no conflict of interest.TS18

*Special issue statement.* This article is part of the special issue "Sub-seasonal to seasonal hydrological forecasting". It is a result of the HEPEX workshop on seasonal hydrological forecasting in Norrköping, Sweden, on 21–23 September 2015.

OR

This article is part of the special issue "Sub-seasonal to seasonal hydrological forecasting". It is not associated with a conference.TS19

*Acknowledgements.* This study was financially supported by the EUPORIAS project (EUropean Provision of Regional Impact Assessment on Seasonal-to-decadal timescale), grant agreement no. 308291, funded by the European Commission (EU) project in the Seventh Framework Programme. Revision of this paper was carried out in the research programme "JPI Climate & Belmont – Climate Predictability" under project number ALWCL.2016.1, funded by the Netherlands Organisation for Scientific Research (NWO).

Edited by: Ilias Pechlivanidis
Reviewed by: three anonymous referees

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

## Remarks from the language copy-editor

**CE3**     Please note that, in regard to the "hindcast" abbreviations, the plural "s" should be added to the abbreviations when you refer to "hindcasts". In some instances, namely those in which you used a plural verb or otherwise seemed to be referring to the plural form, the "s" was added. In other instance, such as when definite or indefinite articles were used, I assumed that you were referring to the singular "hindcast". Please check the accuracy of these abbreviations throughout the manuscript and suggest changes where necessary.

## Remarks from the typesetter

**TS11**     Please provide a statement on how your underlying research data can be accessed. If the data are not publicly accessible, a detailed explanation of why this is the case is required. The best way to provide access to data is by depositing them (as well as related metadata) in reliable public data repositories, assigning digital object identifiers (DOIs), and properly citing data sets as individual contributions. Please indicate if different data sets are deposited in different repositories or if data from a third party were used. If no DOI is available, assets can be linked through persistent URLs to the data set itself (not to the repositories' home page). This is not seen as best practice and the persistence of the URL must be secured.

TS21     Please provide more information.

TS22     Please provide all author names.

TS23     Please add page range or DOI.

TS24     Please add page range or article number with DOI.

TS25     Please add page range or article number with DOI.

TS26     Not used in the text.

TS27     Please add location.