# Peer review of "Seasonal hydro-meteorological forecasts for Europe: sources of skill Wouter Greuell, Wietse H. P. Franssen and Ronald W. A. Hutjes Wageningen University and Research all authors: Water Systems and Global Change (WSG) group, Wageningen University and Research, Droevendaalsesteeg 3, NL 6708 PB Wage"

_Hydrology and Earth System Sciences, 2016_

## Referee Comment (RC1) · Anonymous Referee #1 · 30 Dec 2016

**Review of "Seasonal streamflow forecasts for Europe – II. Explanation of the skill" by W. Greuell et al.**

**Reviewed:** December 2016

**Recommendation:** The manuscript is acceptable with major revisions.

In this paper, the authors present the sources of skill of a model-based seasonal hydrological forecasting system, which produces hydrological forecasts for up to seven months of lead time over Europe. Seasonal hydrological forecast systems over Europe are scarce, as well as the analysis of the sources of the skill over this region, which makes this work relevant to HESS and to the wider hydro-meteorological community.

The authors analyse the sources of skill in the seasonal meteorological, discharge, runoff and evapotranspiration forecasts, using a variety of skill metrics and experiments. This complete and very interesting analysis enables to disentangle the relative contributions of initial hydrological conditions of soil moisture and snow and of meteorological forcing on the skill of seasonal hydrological forecasts up to several months of lead time. The results would however largely benefit from being more concise and structured, in order to guide the readers throughout the paper. To this end and for reproducibility, some results would also need more explaining in the methods section of the paper (i.e., climate change).

I would like to raise two major comments about the methods of this paper, which should be addressed by the authors. These are:

1. Presenting the skill of the meteorological forecasts is a great idea as it highlights the importance of meteorological forecasts for seasonal hydrological forecasting. Furthermore, it shows the differences between the skill in seasonal meteorological and hydrological forecasting, which is a prime motivation for producing hydrological forecasts at seasonal time scales. Nevertheless, I do not fully understand the choice of presenting the skill of the raw S4 forecasts rather than the bias-corrected S4 forecasts, ultimately used to produce the hydrological forecasts in WUSHP. Therefore, I believe that it would make more sense if the skill of the bias-corrected S4 forecasts was presented in the results of the paper, with appropriate skill metrics. Otherwise, readers might question your choice to bias-correct the S4 forecasts in WUSHP.

2. The ESP-experiments carried out for this paper are very original! However, I am not sure if I agree with the way that you designed your ESP-experiments (ESPall, ESPsoilm, ESPsnow). The use of an identical meteorological forcing resampled from the S4 hindcasts for each year might in the end produce an artificial signal, which might lead to a biased analysis of the skill in this paper and is not the aim of the experiment design. Please read and address the detailed comments I have made about this in the rest of the revision. Additionally, the reasons you give for designing all of the experiments differently from the widely used standard ESP and reverse-ESP are not sufficient to argue your choice. In the methods, you state that it is in order for these experiments to be closer to the Full Hindcasts, which would not be the case when using the standard ESP and reverse-ESP formulations. However, I do not see the need for the experiments to be close to the Full Hindcasts, at least in the context of this paper. After reading the detailed comments I have made in this revision referring to the ESP-experiments, please consider either arguing more thoroughly why you have decided

to take a different approach from the standard ESP and reverse-ESP or changing the experiment designs.

The paper is overall written in a generally fluent and precise language. As a whole, I thought that this paper provided valuable results and I would therefore be pleased to see it published in HESS, after major revisions. Below are comments which will hopefully help the authors to improve the paper and highlight the value of the results it contains.

**Title:** The title is pertinent with regards to the contents of the paper. However, the formulation "Explanation of the skill" could be rephrased in order to sound more scientific. You could consider rephrasing it to for example "Sources of the skill".

**Abstract:** Overall, the abstract provides a complete summary of the paper. It could however benefit from being more concise; here are a few suggestions in that direction:

- Page 1, line 12: could you please consider rephrasing "hindcast simulations […] were carried out" to "hindcasts […] were generated"? The term simulations could be confusing here as you are referring to forecasts.
- Page 1, line 17: please change "simulations" to "hindcasts" for the two instances.
- Page 1, line 20-21: this sentence could be removed from the abstract, not hindering the content of the abstract and making it more concise overall. Please consider doing so.
- Page 1, line 23-24: the sentence could be shortened and thus made more concise by removing "to all potential sources of skill but".

**Introduction:** The introduction is interesting and introducing this paper's content with a summary of the previous paper's main findings is a great idea. The introduction however contains a lot of overlap in what is being said. Here are a few suggestions that could maybe help to make the introduction more concise and structured:

- Page 1, line 26: the word "may" sounds like society may also not benefit from such forecasts. It would therefore be interesting to refer to papers tackling this topic, such as: Viel et al. (2016), Soares and Dessai (2016), Crochemore et al. (2016), among others.
- Page 1, line 28: it would be good to add references for other applications of the seasonal predictions, as done for the energy generation sector.
- Page 1, lines 30-31: please consider rephrasing the beginning of the sentence to "WUSHP produces hydrological simulations and forecasts from the Variable Infiltration Capacity […]".
- Page 2, line 1: could you please rewrite "[…] in runoff is fading […]" to "[…] in runoff was found to be fading […]"?
- Page 2, lines 1-2: please consider rephrasing this to "[…], but some significant skill remained up to 7 months of lead time".
- Page 2, line 3: could you please change the word "causes" to "sources"? which is widely used in this context.
- Page 2, line 3: please consider changing "[…] along two lines" to "[…] and is structured in two main parts". Then in the following paragraphs introduce the two parts by saying something similar to: "First, an analysis of the skill […] is carried out" and "In a second part, sources of predictability are analysed […]" (page 2, line 12).
- Page 2, line 4: could you please specify here what variables of the S4 meteorological forcing are analysed in this paper?
- Page 2, line 7: "starting date" or "initialisation month" is better here than "start period".

- Page 2, lines 8-11: these few lines sound too much like results, it should sound more like a literature review. Please consider rephrasing these sentences to sound more like an introduction material. Referring to specific maps of the paper is for instance not adequate here.
- Page 2, lines 14-17: could you please consider combining this part of the introduction with page 2, lines 23-24? which is essentially a repetition of the former.
- Page 2, line 17: please add the word "contributions" after "soil moisture and snow initial conditions".
- Page 2, line 18: the ESP refers to a forecasting technique rather than modelling.
- Page 2, line 19: instead of "as realistic as possible and vary from year to year", a clearer formulation would be for example "our best estimates of the current initial conditions for this specific forecast starting date".
- Page 2, lines 26, 30, 38 and 39: please change "simulations" to "forecasts" or "hindcasts".
- Page 2, line 26: consider rephrasing "is as realistic as possible and is" to "it is our best estimate of the current meteorological conditions,".
- Page 2, lines 32-33: I would rather use the term "climatological information" instead of "no information at all", which is not accurate, and then ", which is the case when they have a climatological distribution" could be removed.
- Page 2, lines 35-36: the sentence "All of these studies basically looked at uncertainty in seasonal forecasts." could be removed as it is not necessary and does not sound so good.
- Page 2, lines 37-38: rephrase this to "(2010), we will first look at the skill of the ESP hindcasts which we will then compare to the standard […]".
- Page 3, line 1: could you please specify here the total skill of what hydrological variables will be quantified by removing one or more sources of skill?
- Page 3, lines 1-3: the sentence starting with "It should be noted" is a repetition of what was already said earlier in the introduction. Could you please consider removing it here?
- Page 3, line 4: could you please rephrase this to "the sources of skill for seasonal hydrological forecasting over Europe"?
- Page 3, line 4: it would be good to specify what is dominated by initial conditions.
- Page 3, lines 13-14: it would be nice if you could add references for this use of evapotranspiration predictions.
- Page 3, line 15: please change "of evapotranspiration" to "in evapotranspiration forecasts".
- Page 3, lines 18-22: could you please specify the sections for each of these different analysis parts? As was done on page 7, lines 10-14.
- Page 3, lines 19 and 20: it is not clear from the introduction what is meant here by "the various ESP experiments". It becomes clearer after reading the methods section though. Could you thus rephrase this here to "the ESP and reverse-ESP experiments"?
- Page 3, line 22: please change "evaporation" to "evapotranspiration" here.
- Page 3, line 22: the sentence about additional figures is not appropriate here. Please consider moving it to the methods or results section of this paper.

**Section 2.1:**

- Page 3, lines 25-29: in this description section, it would be nice if the time step of the simulations, as well as downscaling of the meteorological forcing for the hydrological simulations was mentioned.
- Page 3, line 26: please specify that the bias correction is for the meteorological forcing.

- Page 3, line 27: could you specify hindcasts of which variables are used for this paper? Namely runoff, discharge and evapotranspiration.
- Page 3, line 33: "each of" can be removed here.

**Section 2.2:**

- Page 4, line 6: are those terciles of the observations or of the forecasts? It would be good to specify here.
- Page 4, lines 6-9: I would explain here that for that reason this paper presents results only in terms of the correlation coefficient.
- Page 4, line 16: please specify what is the maximum area that a basin can reach in order to be called a small basin in this paper.
- Page 4, lines 16-18: I would move these results earlier, after the sentence about the comparison between theoretical and actual skill on page 4, lines 12-13, where it fits better.
- Page 4, lines 19-23: I do not understand why you decided to analyse the skill of the non-bias-corrected (raw) S4 forecasts here as you are using the bias-corrected S4 forecasts to produce your hydrological forecasts for this paper. It would thus make more sense to present the skill analysis of the bias-corrected forecasts here. Also, the fact that there are only negligible differences between the bias-corrected and the raw S4 forecasts is, as you mention it, due to your choice of the metrics to compare them. I would thus suggest to use different skill metrics for this specific comparison analysis. If there are still only negligible differences with appropriate skill metrics, it would be ideal to mention that, as previously shown by Wood et al. (2016), negligible meteorological forcing skill improvements can lead to large seasonal streamflow skill improvements (as you mention it in your discussion section), which is why you decide to bias-correct the S4 forecasts here.
- Page 4, line 25: please add "in the scores overview" after "high temporal resolution".
- Page 4, lines 27-28: "lead month zero" is present in many results in this paper, I would thus remove this sentence which is not accurate.
- Page 4, line 29: please specify what will be analysed at the level of the entire domain, the skill?
- Page 4, lines 30-32: please remove the example, sit does not fit here.

**Section 2.3:** for this section, it would be good to make a figure of the various "ESP experiments", this would help the readers understand exactly what was done here.

- Page 4, line 34: you could also refer to the ESP experiments with "ESP" and the reverse-ESP experiment with "reverse-ESP" or "revESP", which would be much clearer. Also, please make sure that you use either the term "ESP experiments" or "ESP-experiments" if you decide to keep this terminology.
- Page 4, line 36-page 5, line 3: ESPall
    o It is not clear to me how the ESPall can have 15 members, since there are 28 years of hindcasts from which the members can be selected. Are some years not used? This should be clarified here.
    o It is mentioned in the results that the same meteorological forcing is used each year (this should be made clearer in the methods). However, I am not sure if this is a good resampling strategy. It could indeed be that the members selected lead to an artificial and persistent skill/signal in certain regions and for some initialisation dates. I would suggest to resample the members for the ESP in a random way for each year individually, the forcing would thus vary for each year of the forecasts.

Alternatively, you could show here that using the same meteorological forcing each year, or using a different meteorological forcing by randomly resampling the members for each year, leads to the same results and that you thus decided to use the former and simpler resampling method.

- o It is in theory a nice idea to resample from the S4 hindcasts instead of the observed meteorological conditions. However, the wider reason for using the standard ESP as a reference to analyse the skill of a seasonal hydrological forecasting system is because it is a computationally cheap method (ideal for operational forecasting) invented when seasonal meteorological forecasts were not skilful enough and based on the assumption that previous years' meteorological conditions are a good indication of future meteorological conditions for the same time of the year. The standard ESP is furthermore known to be a skilful reference and having a more skilful seasonal hydrological forecasting system (here called Full Hindcasts) would guarantee that it is skilful. Here, resampling from the S4 hindcasts is not computationally cheap since you first have to produce those meteorological hindcasts. I also do not entirely understand why you would want this ESP experiment to be as close to the Full Hindcasts as possible. Also, avoiding to reproduce the reference simulation is not a good argument here as this can also be avoided in the standard ESP by simply not selecting the current year. You thus have to argue the choice for this alternative ESP method better in order to use it for your paper. Otherwise, please consider redesigning the experiments. This will impact all other "ESP experiments" of this paper (including the revESP).

- Page 5, lines 4-9: the ESPsoilm and ESPsnow are really clever!

**Section 3.1:**

- Page 5, lines 26-27: please rephrase to "significant skill approaches 5%, the no skill line. Hence […]".
- Page 5, line 28: rephrase to "there is more skill in January, February […] than during the other months".
- Page 5, line 32: please add "(see Fig. 1a)" after "coastal regions".
- Page 5, lines 34-36: this climate change analysis comes as a surprise here, it should be explained in the methods part of the paper to guide the readers throughout the paper. If it is an analysis done in another paper, this paper should refer to it.
- Page 5, line 38: please rephrase to "the theoretical no skill limit".
- Page 6, line 10: specify here that the three summer months are JJA.
- Page 6, line 12: please add "for the summer months" after "function of lead time".
- Page 6, lines 12-13: this is however a completely different area, compared to Europe. Can you really compare the two?
- Page 6, line 30: "mix" is not appropriate here, maybe using "are a combination of" would be better?
- Page 6, lines 36-38: since there is not much to show in the figure and this paper already contains many figures, I would suggest to move the figure to the supplementary material and say that it is not shown here in the text. You could in the text then say what fraction of the domain has skill for "lead month 0".

**Section 3.2:**

- Page 7, line 2: could you please specify here what other sources is referring to? Initial conditions?
- Page 7, lines 3-6: this climate change analysis was not mentioned in the methods, please mention it there.
- Page 7, lines 10-14: I find that this whole paragraph describing the content of the following results breaks the results section. I would remove it or remind the readers of the results structure in the methods rather.

**Section 3.2.1:**

- Page 7, line 18: this is however not entirely true, in the companion paper, some key differences were highlighted between runoff and discharge. Please consider rephrasing this to say that they show a high degree of similarity in terms of magnitude and spatial patterns of skill, or remove the sentence as a whole.
- Page 7, line 20: please specify that the reverse occurs beyond "lead month 1" for most target months, because it does not occur for all.
- Page 7, lines 21-23: there are quite a few differences between the large and the small basins plots (Fig. 4c and Fig. 4d respectively). The differences are however not highlighted here are it is not the main focus of this paragraph. This questions the existence of the two figures. I would either merge small and large basins in one figure, since there is not distinction in the text, or raise the differences (even briefly) in the text.
- Page 7, lines 26-34: this is quite a nice explanation for the reversal of the skill! However, this is hence due to the ESPall experiment design: the use of the same S4 forcing for each year of forecast produced. As discussed in the methods, this should probably be changed to using different random forcing for each year. Because it could be that this specific selection of S4 forcing made here leads to non-random weird skill patterns, in other words to some random signal. This is however not the goal of the method, which is as you said to assess the importance of initial conditions for seasonal discharge and runoff forecasting and the impact of losing the knowledge about the future meteorological forcing, and using random meteorological forcing from previous years of hindcasts as proxy for future meteorological forcing, on the seasonal runoff/discharge forecasting skill.
- Page 7, lines 35-36: it would be interesting to know whether for specific regions in Europe the revESP is more or as skilful as the ESPall for certain target months-lead times combinations. Could this be done and added here?
- Page 7, lines 38-39: the parentheses content is not needed here, the readers can go back to this part of the results if they want to read the specific numbers and it breaks this part of the results. These numbers were however not stated in section 3.1 and should be moved there.
- Page 8, lines 1-4: this is a very interesting observation!

**Section 3.2.2:**

- Page 8, lines 6-7: this is true compared to the ESPsnow experiment and should be specified.
- Page 8, lines 8-9: is this however true for all lead times?
- Page 8, line 14: please, first say what the figure shows in general.
- Page 8, lines 14-18: could you please specify which ESP experiment (ESPsoilm, ESPsnow or ESPall) you are referring to when you write those results, it will help the readers to understand them faster.
- Page 8, line 17-18: I am not sure what is meant by "combined initialisation map". Please rephrase.

- Page 8, lines 19-22: I would move this section earlier, when you are talking about Figure 5, to make it more structured. You can then refer back to this feature when looking at the maps of Figure 6.
- Page 8, lines 25-26: it is hard to understand this point, please consider rephrasing.
- Page 8, lines 27-30: this is a very interesting observation!
- Page 9, lines 2-9: this is a very interesting observation, it would be interesting to show the ESPsnow for soil moisture for May with lead 0, for a comparison. So it is also proving that spin-up is important for hydrological modelling.
- Page 9, lines 6-9: this is a repetition of page 9, lines 2-6. Please consider removing this repetition or combining both explanations.
- Page 9, lines 10-12: this paragraph should be moved earlier, when Figure 5 is described, which would make it more structured and hence clearer to read.
- Page 9, line 12: this order rather depends on the month, not the season, because you are talking in this paper in terms of months. Please change.
- Page 9, lines 13-15: could you please state here what was done exactly with those maps? Or maybe say this in the methods section.
- Page 9, lines 13-19: so none of these hotspots have skill thanks to the meteorological forcing?

**Section 3.3:**

- Page 9, line 21: I would repeat here the intrinsic value of evapotranspiration hindcasts.
- Page 9, lines 21-22: "the power […] ESP experiments" is an odd phrase, rephrase or remove.
- Page 9, lines 23-24: I would remove the piece of the sentence about the April and July decomposition. It is not needed and distracts the readers from the first analysis.
- Page 9, line 25: please explain the overall Fig. 9a before entering into details.
- Page 9, line 25: specify that the levels of predictability in Fig. 9a are for the Full Hindcasts.
- Page 9, line 27: could you please remind the readers in between parentheses what the three ESP experiments are.
- Page 9, line 29: specify that you are talking about the evapotranspiration hindcasts.
- Page 9, lines 33-34: in this sentence, add in between parentheses the ESP experiment you are talking about (revESP, ESPsoilm, ESPsnow), to guide the readers through the results nicely.
- Page 10, lines 1-2: this is during the snowmelt season, please mention.
- Page 10, lines 3-10: this part of the results will benefit greatly from explaining the climate change analysis in the methods section of the paper.
- Page 10, line 11: remind the readers the skill of what variable they are currently looking at.
- Page 10, line 20-21: specify that this is for evapotranspiration hindcasts in April.
- Page 10, lines 21-26: the fact that both temperature and evapotranspiration hindcasts may have the same predictability source is a hypothesis here. Please rephrase the sentences to sound like one.
- Page 10, lines 29: please remind the readers once again what three ESP experiments you are referring to.
- Page 10, lines 30-31: please consider saying that this result can be drawn from the fact that the ESPsoilm shows a higher skill than the ESPsnow and revESP for the Mediterranean.
- Page 10, lines 34-35: please specify which ESP experiments those are.
- Page 11, line 4: is Figure 11f really needed? There are already a lot of figures so I would consider removing it.

**Discussion:**

- Page 11, line 8: please use the terms runoff or discharge rather than streamflow here, to be consistent with the rest of the paper.
- Page 11, line 13: it is not really clear what you are referring to when you say the "uncertainty strategy", could you please rephrase?
- Page 11, line 23: remind the readers that these hotspots regions and periods of skill were identified in the companion paper.
- Page 11, lines 24 and 27: the term "sources of skill" is preferred over "causes of skill".
- Page 11, line 28-page 12, line 8: this analysis is very interesting!
- Page 12, line 10: instead of "replaces" I would say "becomes less skilful than […]".
- Page 12, lines 11-13: you can however not exactly compare your results to results from other papers here as their ESP experiment was different from yours I suppose. You could still cite their results to compare to yours but make sure to highlight this difference.

**Conclusions:** the conclusion is overall too long. I would shorten it to keep only the main results of the paper. Here are some suggestions:

- Overall, please don't refer to figures here.
- Page 13, line 11: describe quickly the different ESP experiments.
- Page 13, lines 14-15: when you say "other ESP-experiments" mention that these were performed in this paper. Otherwise it sounds like you are talking about ESP experiments from other papers.
- Page 13, lines 18-19: I would remove the piece of sentence "Similar […] domain".
- Page 13, lines 22-29: I would summarise this whole paragraph in just a few sentences. Just to convey the main conclusion from these results. The detailed results that you are currently describing can be found in the results section of the paper.
- Page 13, lines 30-36: same as above.
- Page 13, line 37-page 14, line 5: I would move this paragraph in the discussion section of the paper.

**Figure 1:**

- Please consider swapping Figure 1a and 1b as you are first describing result from Figure 1b in the results.
- Figure 1a: could you please add a label for the colour bar saying that this shows R?
- Figure 1b:
    - Please consider making the y-axis a log scale so that we can see what is happening around the 5% line?
    - Would making a colour scale for the initialisation months be possible? This would maybe make the plot more understandable.
- Caption:
    - Please specify that the legend which provides the percentage of cells with significant R values is in the top left corner of Figure 1a.
    - Could you also state that darker red colours signify a better skill?

**Figure 2:**

- Figures 2a and 2b:

- These figures look quite messy for lead times 1 and 2. Would using a log scale for the y-axis help with that?
- Also consider making a colour bar for the different starting months, as suggested for Figure 1a.
- Figure 2c: the labels of this figure are quite messy. Could you please put a legend outside the figure instead?
- Please remove the general title, it is already said in the caption.
- Specify that the colour bar is for R by adding a label next to it.
- Caption:
  - Instead of writing "As Fig. 1" I would mention here again what this figure is. Because it is easier to read directly under the figure than having to jump from a figure caption to the other figure.
  - Consider removing the exclamation mark after "not the trend itself".

**Figure 3:** I would suggest to remove this figure and put it in the supplementary material. In any case, all comments made to Figure 1b apply here, as well as the caption explanation instead of writing "As Fig. 1b".

**Figure 4:**

- These figures are too messy, please make a common legend to explain what the different lines are.
- Add an x-axis label specifying that these are target months.
- You could remove the x-axis tick labels for Figure 4b as it is shown in the Figure 4d below.
- You could remove the y-axis tick labels for Figures 4b and 4d as they are the same as in the Figures 4a and 4c.
- Caption:
  - Please explain what the figures show in the caption instead of writing "As Fig. 2c".
  - Instead of "first two panels" say "top two panels" and instead of "other two panels" say "bottom two panels".

**Figure 5:**

- Please remove the title as it is specified in the caption already.
- Could you please make a legend for the different lines, it is currently quite messy?
- Add an x-axis label specifying that these are target months.
- Explain what the figures show in the caption instead of writing "As Fig. 4".

**Figure 6:**

- Consider removing the main title.
- Please add a label for the colour bar.
- Caption:
  - Explain what the three ESP experiments are.
  - Could you also explain what the figures show in the caption instead of writing "For more explanation, see Fig. 1a"?

**Figure 7:** same comments as for Figure 6, except the comment regarding the different ESP experiments.

**Figure 8:** same comments as for Figure 7, except the comment about the main title. Additionally, the caption should not describe the results.

**Figure 9:**

- Consider removing the main title.
- Figure 9a: same comments as for Figures 1b, 2a and b.
- Figure 9b: same comments as for Figure 5.
- Figure 9c: same comments as for Figures 2c and 4a, b, c and d.
- Caption:
  - Explain what the figures show in the caption instead of writing "for more explanation, see Fig. 1b".
  - Please specify what the different ESP-experiments are.

**Figure 10:**

- Please add a label for the colour bar.
- Specify what the figures show instead of saying "for more explanation, see Fig. 1a".

**Figure 11:**

- Consider removing the main title.
- Add a label for the colour bar.
- Caption:
  - Could you specify what the figures show instead of saying "for more explanation, see Fig. 1a"?
  - "The final is panel f and depicts the skill […]".
- Consider removing Figure 11f.

**Figure 12:**

- You do not need a colour bar for each sub figures: Figures 12a and b can share one, and Figures 12c and d as well.
- Add the labels for the two different colour bars.
- Caption:
  - The caption describes the results, it should not.
  - Please specify what the figures show instead of saying "for more explanation, see Fig.s 1".

**Technical corrections:**

- General:
  - Could you please add the word "meteorological" in front of "forcing" when you refer to meteorological forcing. It will make it clearer to the readers what you are talking about.
  - Please consider changing "lead month" to "month of lead time" or "lead time", which is more widely used, and will hence be clearer for the readers even without having read the methods section.
  - Could you please replace "panel" with Fig. figure# subfigure#? E.g., for Figure 5, panel c would be replaced by Fig. 5c.
  - Could you please consider renaming the terms "pseudo-observations" and "real observations"? I would for example use "analysis" (as done in meteorology) or "simulations", for the pseudo-observations, and simply "observations" for the "real observations".

- o Could you please change "North" to "Northern", "South" to "Southern", "West" to "Western" and "East" to "Eastern" when in front of a country's name?
- Page 1, line 10: could you please add a comma after "In WUSHP"?
- Page 1, line 12: could you please change "To explain skill" to "To explain the skill"?
- Page 1, line 13: please consider using the term "analysed", or something more scientific sounding instead of "looked at".
- Page 1, line 13: please change "of the first […]" to "for the first […]".
- Page 1, line 14: instead of "later", consider using the word "subsequent".
- Page 1, line 14: "Seasonal forecasts of temperature".
- Page 1, line 30: consider removing "that was" and adding a comma before "built […]".
- Page 1, line 33; page 2, line 3: please change the "[…] and lack of skill […]" to "[…] or lack thereof […]".
- Page 2, line 4: please add a comma in "For S4, this was done […]".
- Page 2, line 5: rephrase "with initialisation at the" to "initialisation on the".
- Page 2, line 16: rephrase "and separated" to "to separate".
- Page 2, line 17: remove the second dot.
- Page 2, line 30: remove the comma after "(2008)".
- Page 2, line 35: please change the sentence to "changes in the information of the meteorological forcing and the initial conditions.".
- Page 2, line 37: the word "However," can be removed.
- Page 3, line 4: add "the" in front of "sources".
- Page 3, lines 10-11: move "also" to before "play a role".
- Page 3, line 12: add "the" in front of "skill".
- Page 3, line 16: "Thus" is not needed here.
- Page 3, line 29: change to "so a total of 5400 simulations (30 years * 12 months * 15 members) was carried out.".
- Page 3, lines 31-32: consider changing one "namely" to a synonym.
- Page 4, line 5: please add "the" in front of "Relative".
- Page 4, line 7: change "are similar" to "were similar".
- Page 4, line 9: if you put capital letters for Below Normal and Above Normal please also add the abbreviation in parentheses after the term is introduced.
- Page 4, line 15: is the hyphen needed in between "large" and "basins"?
- Page 4, line 16: the quotation marks are not needed around the word "observations" here.
- Page 4, line 24: add "a" in front of "relatively".
- Page 4, line 19: please add a comma after "Here".
- Page 4, line 26: please add a comma after "(2005)".
- Page 4, line 29: please add a comma after "result sections".
- Page 4, line 29: please consider changing the term "remarkable" to "outstanding" or "noteworthy".
- Page 4, line 30: please add the word "intend to" in front of "provide".
- Page 4, line 34: please add a comma after "total".
- Page 5, line 23: please write "are used here as a reference".
- Page 5, line 24: "A summary of the skill".
- Page 5, lines 24-25: "in Fig. 1a with statistically".
- Page 5, line 29: rephrase to "target months (not shown here) hot spots […]".
- Page 7, lines 3-4: "the question of how much […]".

- Page 7, line 33: remove the "at" in front of "some time".
- Page 9, line 22: please add a comma after "First".
- Page 9, line 27: add "the" in front of "three ESP".
- Page 10, line 14: add a comma after "in revESP".
- Page 10, line 14: add "the" before "revESP", for both instances.
- Page 10, line 30: "From the ESP experiments, it can be concluded […]".
- Page 11, line 4: there is a space missing between "July 1" and "(panel f)".
- Page 11, line 22: "The same is probably true for the S4 hindcasts".
- Page 12, line 8: replace "less than" with "lower than".
- Page 12, line 19: add "the" in front of "ESP".
- Page 12, line 33: please add "for" in front of "practical".
- Page 12, line 36: add "also" in front of "demonstrates".
- Page 13, line 17: please add a comma after "melt season".

---

## Referee Comment (RC2) · Anonymous Referee #2 · 15 Jan 2017

This is Part II of a paper describing a new dynamical ensemble seasonal streamflow forecasting system for Europe, which uses meteorological forcing from a coupled prediction system in the VIC hydrological model.

Dynamical continental scale ensemble prediction systems are at the cutting edge of seasonal streamflow forecasting. The general aim of explaining the sources of predictability in different regions is interesting and worthwhile, and traditional ESP and reverse ESP methods are well-established techniques to do this. The paper is generally clearly written and I acknowledge the considerable effort the authors have put into producing this paper. In short, I think the forecasting system is interesting, as is the aim of investigating of sources of skill, and that it deserves ultimately to be published. However, in my view the metrics/methods used to assess prediction performance are too rudimentary, to the point where it is difficult to understand how the system performs. I also had some reservations about their attribution of skill to climate change, the revESP method used here, the description of ESP. Accordingly, I believe the paper requires major revisions before it can be published.

General Comments

Some of my objections relate to both parts I and II of this paper (as part II relies heavily on part I), so the authors may wish to address them in both (or either) papers. Specifically, my major objections are:

1) The authors essentially rely on correlation between forecasts and observations as the major metric of performance. In my view they should not, but should use RPSS instead. The argument (made in the Part I paper) that correlations are 'easier to understand' than RPSS simply doesn't hold water in my opinion: skill scores that describe performance in relation to climatology (like RPSS) make it much easier to understand the value of the forecasting system (even against the 'pseudo observations' used in this paper) than correlations.

In addition, I do not agree with the authors' contention that the RPSS and correlations "are similar to a high degree". The theoretical differences between skill scores and correlations have been documented by Murphy (1988), who concluded: "...use of the correlation coefficient (or its square) may lead to substantial overestimation of forecasting performance" and that "...it is more appropriate to interpret the square of the correlation coefficient as a measure of potential skill than as a measure of actual skill" These differences appear to manifest in practice for WUSHP. As far as I can see, the only evidence the authors present to demonstrate that correlations and RPSS are similar for WUSHP is Figure 8 in the Part I paper (it shows forecasts for May at lead 2). I could not follow the method used to calculate the statistical significance of the RPSS values (please supply more details), but on the face of it the drop in significant performance from 76% of all grid cells (correlation) to 47% of all grid cells (RPSS) reported by the authors is substantial (i.e., 29% of cells appear to have changed from

being designated as 'skillful' to not skillful). The heat maps in Figure 8 of the Part I paper also show substantial divergences between correlation and RPSS. For example, RPSS values of less than zero skill are shown over much of Poland/Belarus/Ukraine, but this region exhibits high (and significant) correlations. Similar divergences between RPSS and correlations happen over Ireland, southern Spain, much of nothern Africa, eastern Germany, Greece and the Balkans, and substantial tracts of Italy, Romania and western Russia.

Given these differences, and the theoretical preferability of RPSS, I think the authors should replace correlations with RPSS as the major metric for skill throughout the paper (though note the comment #9 in 'Other Comments' about reference forecasts), and change their interpretations/conclusions accordingly. I also recommend that the authors use the word 'skill' in the sense more commonly (though admittedly not universally) used in the forecasting literature - i.e., skill is performance with respect to a reference forecast - rather than as a more general synonym for 'accuracy'.

2) The authors present an ensemble forecasting system without any explicit analysis of reliability in either the Part I or Part II paper. Reliability is a crucial property of any ensemble forecasting system (e.g., Mason and Stephenson 2008; Raftery 2016; among many others). The authors should quantify and discuss the reliability of their forecasting system, using established diagnostics of reliability (I particularly recommend the probability integral transform - see, e.g., Gneiting and Katzfuss (2014) - but attributes/reliability diagrams (Hsu & Murphy 1986) are also suitable for binary forecasts). The authors may choose to address this issue in the Part I paper, but it must be addressed somewhere.

3) No mention is made of cross-validation. Part I alludes to the need to calibrate the VIC model, and quantile mapping is applied. In addition, ESP experiments sample from different years (or they should - see comments #5 and #6, below). All these need to be robustly cross-validated to ensure forecast performance is not overstated (e.g., using leave-x-year out cross-validation). Please describe the cross-validation methods

employed.

4) In a number of instances the authors ascribe (or do not ascribe) fractions of skill to climate change by examining trends in data. There are a couple of issues here. First, the methods for detrending/tests of statistical significance of trends are not explained - so I do not know what is being detrended or how (in an Appendix is fine). Second, to do this analysis the authors assume that trends in data are somehow causally related to forecast skill (implicit in Figure 2 f-i, and their discussion of these figs). It is not clear to me why this should be so in all the cases presented - particularly for climate variables like temperature, about which the authors state "Skill in summer temperature is related to climate change occurring in both the observations and the hindcasts...". Because S4 forecasts are initialised by assimilating observations, it's reasonable to expect that the hindcasts have trends in them (induced by the initialisation) that are similar to observations. So it seems (I think) the authors are implying that the thermodynamical/dynamical responses of S4 to initial conditions may not reflect thermodynamical/dynamical changes induced by climate change. Again, I do not know why this would be (after all, climate models are used routinely for future projections, so they are assumed to work similarly in future as now). Please first explain how trends could impact predictability before drawing any conclusions on how climate change driven trends in data influence skill.

5) I am not very familiar with the VIC model, but I would assume that some of the internal states could be highly correlated/anticorrelated (as in most hydrological models). Using states averaged from different periods - as done here for the revESP method - could destroy these correlations, and could lead to unrealistically poor simulations. What I think the authors should have done is generated an ensemble of states by forcing VIC with resamples observations from a number of years (as in the original revESP) and then forced each of these with the ensemble of climate forecasts. This would lead to more ensemble members (15* number of years sampled), but would allow much more satisfacory diagnosis of the contribution of meteorological forcing to overall skill.

This means the revESP experiments would have more ensemble members than the other experiments, but I think the benefits of this approach outweigh potential artefacts arising from different ensemble sizes.

6) The method for the ESP is described as follows: "We did not select atmospheric forcings from observations (e.g. WFDEI), which is the strategy employed in most published ESP experiments. By selecting the forcing from the S4 hindcasts, the ESP experiments remain as close as possible to the Full Hindcasts." I assume the authors have used some way of sampling from different years, but the way this is written makes it possible to interpres this as if the ESP forecasts could simply be analogous to hindcasts. (If the authors have used 'ESP' in the latter way, 1) they should not call it ESP and 2) they cannot claim to isolate the source of skill, which is their aim.) The authors need to clarify what they have done here. If, as I've assumed, they have sampled from hindcasts, this is essentially a new method (which is a good thing). In this case the authors need to spell out their method clearly, including sampling strategies for ensemble members, etc.. They may also like to give it a new name - e.g. HESP for 'hindcast ESP', or similar.

7) As the authors are introducing a new system, it needs to be put in the context of existing operational and experimental prediction systems. The introduction does not really do this at present. For example, the authors could note that statistical systems can much more easily be configured to produce reliable ensemble forecasts (e.g. Madadgar and Moradkhani 2013; Wang and Robertson 2011). In addition, other experimental dynamical forecast systems have attempted to explicitly deal with problems related to reliability (Yuan 2016; Bennett et al. 2016). (While Yuan (2016) is mentioned, the authors do not note a crucial difference between this system and WHUSP - that is, Yuan's hydrological post-processing step that attempts to ensure reliable ensembles.) A paragraph explaining how the WHUSP approach compares to existing systems, including any differences in the aims of WHUSP compared to other systems, would be a useful addition to the introduction. (For example, it is fine to validate against pseudoreality if this is how the system is to be used in operation, but if the aim of WHUSP is to predict actual inflows to reservoirs then it should be validated against observations.)

8) Quantile mapping has several limitations as a method for post-processing ensemble forecasts - in particular that it does not correct for errors in reliability because it ignores information that is available from correlations between hindcasts and observations (see Wood and Schaake 2008; Madadgar et al. 2014). A statistical calibration (e.g. Gneiting et al. 2005; Schepen et al., 2014) is probably preferable. This should be acknowledged somewhere.

9) It was not clear to me what was used as the reference forecast when RPSS was calculated. I suggest a cross-validated measure of climatology that varies with month (e.g. an ensemble of resampled historical streamflow, or similar). Please clarify.

Specific (minor) comments

Line 18 'The term ESP refers to...hindcasts'. Not only hindcasts - ESP systems are widely used to produce forecasts

Line 20 'reference simulation'. As noted above, 'reference' forecasts in forecasting literature are frequently used to denote a benchmark for performance. I suggest using a different term than 'reference simulation' here, because it is not really being used as a reference. Suggest simply 'simulation'

Line 21 'identical'. ESP experiments are often cross-validated (depending on the aims of the experiment), so forcings may not be 'identical'.

Line 21 'This is not surprising'. It's not surprising if you consider correlation as synonymous with skill. As I argue above, I don't believe this is justified. As Murphy (1988) points out, skill scores have components that consider, e.g. conditional and unconditional biases, which correlations ignore. Clearly bias correction will influence these aspects of skill.

Lines 1-2 'By selecting the forcing from S4 hindcasts, the ESP experiments remain as close as possible to the Full Hindcasts'. I do not understand what the authors mean by 'select' here. It could imply that the authors simply used S4 forecasts (i.e. the same as the 'Full Hindcasts'), but this does not make sense (see comment #5). Please clarify.

Lines 14-15 '...which is important since ensemble size affects skill metrics'. I would say what's of more concern is that a small ensemble of 15 members is likely to mean that your skill metrics are subject to considerable sampling uncertainty.

Line 5 '...is linked to climate change...' change to '...could be linked to climate change...'

Line 5 '...by detrending the data...' A brief summary of what is being detrended, the detrending technique and the trend significance test is needed (either a description, which could go in an appendix, or a reference)

Line 6 '...is insignificant across most of the domain...' I assume the trends were analysed only for the hindcast periods. Please note this somewhere

Line 35 '(revESP) always causes much less significant skill' I accept that this will probably be true, but could this result be partly caused by the way in which that the model has been initialised? (i.e with states that may not be correlated - see comment #4)

Lines 1-3 'We explain the enhanced skill in runoff by an indirect effect of the skill of the precipitation forcing in the first lead month, which gradually adds some skill to the model states of soil moisture and snow.' I undestand what the authors are getting at here, but I think this is poorly phrased. The forcing doesn't really 'add skill' to the

model states. It's simpler to say that runoff forecasts are generally more skillful than climate forecasts because they aggregate skill from initial conditions and skill from meteorological forcings.

Line 31-32 '... where the first skill occurs...' What is meant by 'first skill'?

Line 20 'skill in evapotranspiration' - This section is somewhat out of character with the first part of the paper. I am not suggesting it is not important work, but it perhaps would have been better in its own paper.

Line 21 '...hindcasts of evapotranspiration have intrinsic value...' suggest '...hindcasts of evapotranspiration have indepedent of streamflow forecasts...'. Also, please provide a brief summary somewhere of how VIC calculates ET.

P 10

Line 24-26 'Initial conditions of snow and/or soil conditions lead to skill in the temperature hindcasts of the climate model (S4) and initial conditions of snow and soil moisture lead to skill in the evapotranspiration hindcasts of the hydrological model (VIC).' Is there generally agreement in the VIC snow/soil moisture states and the (I presume) observations assimilated by S4?

P11

Line 18 '...their semi statistical forcing is more skilful than the S4 forcing...'. Is is also possible that your hydrological model is more efficient, thereby giving you relatively more skill from initial conditions?

Line 28 '...to what extent they are due to a lack of interannual variability in the processes that eliminate the skill?' As discussed in comment #1, this would be straightforward to answer if you used skill scores calculated against a suitable climatological reference forecast (comment #9)

Line 32 'which is an important skill-eliminating factor'. This won't show up in correlations.

P12

Line 25 'The logical answer is "yes" but such a strategy should then be reconsidered regularly'. As noted in comment #8, statistical calibration methods are available to post-process climate outputs. One of the benefits of these methods is that in the absence of demonstrable forecast skill, they return climatology forecasts. So a more effective alternative than using two forecasting systems might be to use calibrated S4 forecasts as forcing, which would then effectively give an ESP-like forecast when skill isn't there. See Peng et al. (2016) for an example of this applied to S4.

Line 36 'This study demonstrates the power of using pseudo-observations for verification.' I agree, but the usefulness of pseudo-reality also depends on the ultimate aims of the forecasting system. If the aim of the system is to give accurate streamflow forecasts where quantities matter (e.g., forecasting inflows to reservoirs), then the system must be verified accordingly (i.e., against gauged streamflows). I think the authors should acknowledge this.

Typos/grammar

Line 6 delete 'among others'

Line 37 '...of the an...' delete 'the'

Line 27-28 '...snow stops to be available...' change to 'snow is not available'

Line 34 delete 'quite'

Line 11 '...April as...' make it '...April at...'

Line 30 change '...Mediterranean in due...' to '...Mediterranean is due...'

Line 5-6 '...two hotspot region.' Make it 'regions'

Line 33 '...not the case practical applications.' Add 'in'

References

Bennett, J. C., Q. J. Wang, M. Li, D. E. Robertson, and A. Schepen (2016), Reliable long-range ensemble streamflow forecasts: Combining calibrated climate forecasts with a conceptual runoff model and a staged error model, Water Resources Research, 52, 8238–8259, doi: 10.1002/2016wr019193.

Gneiting, T., A. E. Raftery, A. H. Westveld, and T. Goldman (2005), Calibrated probabilistic forecasting using ensemble model output statistics and minimum CRPS estimation, Monthly Weather Review, 133(5), 1098-1118, doi: 10.1175/mwr2904.1.

Madadgar, S., and H. Moradkhani (2013), A Bayesian Framework for Probabilistic Seasonal Drought Forecasting, Journal of Hydrometeorology, 14(6), 1685-1705, doi: 10.1175/jhm-d-13-010.1.

Madadgar, S., H. Moradkhani, and D. Garen (2014), Towards improved post-processing of hydrologic forecast ensembles, Hydrological Processes, 28(1), 104-122, doi: 10.1002/hyp.9562.

Murphy, A. H. (1988), Skill Scores Based on the Mean Square Error and Their Relationships to the Correlation Coefficient, Monthly Weather Review, 116(12), 2417-2424, doi: 10.1175/1520-0493(1988)116<2417:ssbotm>2.0.co;2.

[Figure]

Peng, Z., Q. J. Wang, J. C. Bennett, A. Schepen, F. Pappenberger, P. Pokhrel, and Z. Wang (2014), Statistical calibration and bridging of ECMWF System4 outputs for forecasting seasonal precipitation over China, Journal of Geophysical Research (Atmospheres), 119, 7116–7135, doi: 10.1002/2013JD021162.

Raftery, A. E. (2016), Use and communication of probabilistic forecasts, Statistical Analysis and Data Mining: The ASA Data Science Journal, 9(6), 397-410, doi: 10.1002/sam.11302.

Schepen, A., Q. J. Wang, and D. E. Robertson (2014), Seasonal forecasts of Australian rainfall through calibration and bridging of coupled GCM outputs, Monthly Weather Review, 142(5), 1758-1770, doi: 10.1175/mwr-d-13-00248.1.

Wang, Q. J., and D. E. Robertson (2011), Multisite probabilistic forecasting of seasonal flows for streams with zero value occurrences, Water Resources Research, 47, W02546, doi: 10.1029/2010WR009333.

Wood, A. W., and J. C. Schaake (2008), Correcting Errors in Streamflow Forecast Ensemble Mean and Spread, Journal of Hydrometeorology, 9(1), 132-148, doi: 10.1175/2007jhm862.1.

Yuan, X. (2016), An experimental seasonal hydrological forecasting system over the Yellow River basin – PartÂă2: The added value from climate forecast models, Hydrology and Earth System Sciences, 20(6), 2453-2466, doi: 10.5194/hess-20-2453-2016.

---

## Referee Comment (RC3) · Anonymous Referee #3 · 9 Feb 2017

The manuscript is the 2nd of a companion paper addressing streamflow forecast performance across Europe. The methodology is fairly well described and presented, however substantial clarification and justification is necessary in numerous areas. The authors predominantly limit themselves to evaluation of only a few performance metrics, and report many findings for the whole of Europe. The overall contribution contains meritorious aspects, particularly the performance of this dynamical system, however these need to be highlighted and clarified significantly. Also, distinction and improvement between this and prior studies (e.g. Bierkens and van Beek) is not sufficient.

Specific comments: 1. The title indicates seasonal streamflow prediction, yet the paper focuses on Monthly results for streamflow, temperature, and evaporation. Title not entirely indicative of manuscript focus.

2. Given the spatial heterogeneity of Europe, the authors should provide better justification for reporting predominantly spatially lumped results.

3. Auto-regressive effect (streamflow persistence) not explicitly mentioned or discussed. Is it assumed to be (partially) accounted for in initial soil moisture? For most rivers, particularly large rivers, this is a dominant feature.

4. GRDC has discharge stations downstream of reservoirs, where regulations and management of discharge is often evident. But these have not been corrected (or even noted) in the dataset. Europe is full of situations like this. How have those been accounted for?

5. Unclear (no explanation) of what the ratio of actual/theoretical skill means. Clear that they are closer for larger basins (no surprise) but does the fact that both are far from 1 indicate less "realistic" outcomes? Or does this have little bearing on skill metrics (comparing apples to apples.) Please clarify.

6. How do the three "conditions" relate with the pseudo-obs? For example, report R or RPSS between soil moisture and streamflow by grid? Or snow and streamflow? Could assess for at least a sub-set of locations. This would also give insights as to the value added (or not) by VIC.

7. In addition to reporting the % of cells where R is significant, consider also reporting the mean and standard deviation of R in those cells. The number of significant cells does not necessarily represent the quality of the relationship (e.g. % of cells could increase, but mean decrease. . .) And then discuss.

8. RPSS is mentioned early in the study, but results are not presented. Such categorical skill scores are worth exploring.

9. The authors lightly compare their study outputs with others, namely Bierkens and van Beek, indicating lower performance, likely attributable to the latter's use of semi-statistical forcing. While there are still other meritorious aspects to this current contri-

bution, the authors do not adequately discuss the implications of poorer performance. Are there reasons that the proposed methodology is advantageous as compared with others? Should the GloSea5 approach be used in lieu of the one proposed here? More discussion is needed.

10. In the Conclusion, the authors mention the potential improvement of assimilating soil moisture or SWE to VIC. Why was this not performed and analyzed?

11. Challenging to follow train of through in some parts. Could benefit from the writing be tightened up overall - and simplified in some places. Word choice also needs to be improved in many places (e.g. "Fig. 8 analyses [sic] a remarkable feature." Figures cannot analyze. Figures can illustrate.)

---

## Author Comment (AC2) · 14 Apr 2017

We are very happy with the fact that the editor found three anonymous referees to give their highly informed opinions on our paper. We thank all three for their respective efforts to produce such extensive and constructive reviews.

We adopt most of the suggested textual improvements and specified our action to every remark in the annotated report hess-2016-604-RC2-author-reply.pdf1.

RC2 raises concerns regarding 6 technical issues (RC2's numbering): 1. Choice of metric used to present most results. We use R mostly, RC2 prefers the use of RPSS. [....] [ ... discuss merit of ROCSS rather than RPSS... in context of early warning of extremes...] We believe that RPSS underestimates skill as it addresses all percentiles, whereas it is well known that skill for the extreme percentiles is generally higher and

more relevant to the user than skill for the central percentiles. Similarly, we consider ROCSS for AN and/or BN terciles more relevant than for NN (and these AN/BN are nearly always higher as well. Though the spatio temporal patterns of the dynamics of skill are similar between the three metrics R, RPSS and ROCSS, we agree with RC2 that their statistical significance levels are not necessarily similar (generally the fraction of EU with sign. scores decrease from R > ROC-AN/BN > RPSS » ROC-NN, see figure SM2 paper 1). So first we will better explain how significance is computed. Next we will much better illustrate, describe and explain in the main text of paper 1 the spatio temporal differences found, between the various skill metric, adding to its supplementary material also maps for the other skill scores. In paper 2 we believe this is less relevant as this is about (dynamics of) the sources of skill. However, we will check how different fig 4 and 5 of paper 2 become when using RPSS as skill metric, and add these (in the main paper or SM) depending on the outcome of this check.

2. RC2 is the only reviewer that recommends the analysis of reliability in addition to the other metrics already assessed. We will make a basic assessment on the reliability of WUSHP forecasting system, and present its results in paper 1. Depending on its outcome we might come back to some aspects of reliability in paper 2.

3. We do not fully understand this comments as it seems to mix up some different things. E.g., the quantile mapping referred to in paper 1 is applied to the forcing data, and has nothing to do with VIC model calibration. The latter implies calibrating some model parameters using observed stream flows. We are aware some authors (e.g. Crochmore) also apply a 'leave one year out method' to bias correction (i.e. determine BC factors from other 29 years and apply it to year x). Though, there may be some theoretical merits to do it this way, we believe it practical effects are very small, and as a result many studies derive BC factors from the same set of years as to which they are subsequently applied. We are not sure how cross validation applies to other part our methods. E.g. the 'leave year x out' is not relevant to our ESP implementation since we take the forcing from the hindcasts , not from the reanalysis, so they are never identical

between ESP and reference.

4. We will better explain how any trend in the observations (whether it is due to climate change signal is in principle not relevant, so we will stress this aspect less), that is reproduced in the hindcast, will (mathematically, not physically) lead to an increase in skill.

5. and 6. Yes, we do propose and use an alternative form of both rev-ESP and ESP experiments. With our form of the rev-ESP we assess the value of the actual meteo-rological forecast quality; not the uncertainty in the forecasts due to uncertainty in the meteo forcing, which is the goal of many of the published rev-ESP experiments. This while assuming we have no knowledge of Initial Hydrological States. So we set IHCs to values unrelated to actual IHCs, by using climatological means. We agree some correlations between IHC might get lost, and thus decrease skill. But sampling IHCs from pseudo observations (i.e. historical simulations), as in the original will also decrease skill as dry IHCs might be used for meteorologically wet years and vice versa, which is less a problem when using means, so the average difference between actual and used IHC will be smaller. We will better discuss the implications of the alternative rev-ESP and ESP in the discussion sections, and like suggested by RC2 in his comment 6, to prevent confusion we will give it a different name. [see also our reply to RC1 comment #2]

Other issues raised by RC2 are mostly minor and responded to in the annotated pdf.

Please also note the supplement to this comment:
http://www.hydrol-earth-syst-sci-discuss.net/hess-2016-604/hess-2016-604-AC2-supplement.pdf

**Supplement:**

This is Part II of a paper describing a new dynamical ensemble seasonal streamflow forecasting system for Europe, which uses meteorological forcing from a coupled prediction system in the VIC hydrological model.

Dynamical continental scale ensemble prediction systems are at the cutting edge of seasonal streamflow forecasting. The general aim of explaining the sources of predictability in different regions is interesting and worthwhile, and traditional ESP and reverse ESP methods are well-established techniques to do this. The paper is generally clearly written and I acknowledge the considerable effort the authors have put into producing this paper. In short, I think the forecasting system is interesting, as is the aim of investigating of sources of skill, and that it deserves ultimately to be published. However, in my view the metrics/methods used to assess prediction performance are too rudimentary, to the point where it is difficult to understand how the system performs. I also had some reservations about their attribution of skill to climate change, the revESP method used here, the description of ESP. Accordingly, I believe the paper requires major revisions before it can be published.

General Comments

Some of my objections relate to both parts I and II of this paper (as part II relies heavily on part I), so the authors may wish to address them in both (or either) papers. Specifically, my major objections are:

1) The authors essentially rely on correlation between forecasts and observations as the major metric of performance. In my view they should not, but should use RPSS instead. The argument (made in the Part I paper) that correlations are 'easier to understand' than RPSS simply doesn't hold water in my opinion: skill scores that describe performance in relation to climatology (like RPSS) make it much easier to understand the value of the forecasting system (even against the 'pseudo observations' used in this paper) than correlations.

In addition, I do not agree with the authors' contention that the RPSS and correlations "are similar to a high degree". The theoretical differences between skill scores and correlations have been documented by Murphy (1988), who concluded: "...use of the correlation coefficient (or its square) may lead to substantial overestimation of forecasting performance" and that "...it is more appropriate to interpret the square of the correlation coefficient as a measure of potential skill than as a measure of actual skill" These differences appear to manifest in practice for WUSHP. As far as I can see, the only evidence the authors present to demonstrate that correlations and RPSS are similar for WUSHP is Figure 8 in the Part I paper (it shows forecasts for May at lead 2). I could not follow the method used to calculate the statistical significance of the RPSS values (please supply more details), but on the face of it the drop in significant performance from 76% of all grid cells (correlation) to 47% of all grid cells (RPSS) reported by the authors is substantial (i.e., 29% of cells appear to have changed from

being designated as 'skillful' to not skillful). The heat maps in Figure 8 of the Part I paper also show substantial divergences between correlation and RPSS. For example, RPSS values of less than zero skill are shown over much of Poland/Belarus/Ukraine, but this region exhibits high (and significant) correlations. Similar divergences between RPSS and correlations happen over Ireland, southern Spain, much of nothern Africa, eastern Germany, Greece and the Balkans, and substantial tracts of Italy, Romania and western Russia.

Given these differences, and the theoretical preferability of RPSS, I think the authors should replace correlations with RPSS as the major metric for skill throughout the paper (though note the comment #9 in 'Other Comments' about reference forecasts), and change their interpretations/conclusions accordingly. I also recommend that the authors use the word 'skill' in the sense more commonly (though admittedly not universally) used in the forecasting literature - i.e., skill is performance with respect to a reference forecast - rather than as a more general synonym for 'accuracy'.

2) The authors present an ensemble forecasting system without any explicit analysis of reliability in either the Part I or Part II paper. Reliability is a crucial property of any ensemble forecasting system (e.g., Mason and Stephenson 2008; Raftery 2016; among many others). The authors should quantify and discuss the reliability of their forecasting system, using established diagnostics of reliability (I particularly recommend the probability integral transform - see, e.g., Gneiting and Katzfuss (2014) - but attributes/reliability diagrams (Hsu & Murphy 1986) are also suitable for binary forecasts). The authors may choose to address this issue in the Part I paper, but it must be addressed somewhere.

3) No mention is made of cross-validation. Part I alludes to the need to calibrate the VIC model, and quantile mapping is applied. In addition, ESP experiments sample from different years (or they should - see comments #5 and #6, below). All these need to be robustly cross-validated to ensure forecast performance is not overstated (e.g., using leave-one year out cross-validation). Please describe the cross-validation methods

employed.

4) In a number of instances the authors ascribe (or do not ascribe) fractions of skill to climate change by examining trends in data. There are a couple of issues here. First, the methods for detrending/tests of statistical significance of trends are not explained - so I do not know what is being detrended or how (in an Appendix is fine). Second, to do this analysis the authors assume that trends in data are somehow causally related to forecast skill (implicit in Figure 2 f-i, and their discussion of these figs). It is not clear to me why this should be so in all the cases presented - particularly for climate variables like temperature, about which the authors state "Skill in summer temperature is related to climate change occurring in both the observations and the hindcasts...". Because S4 forecasts are initialised by assimilating observations, it's reasonable to expect that the hindcasts have trends in them (induced by the initialisation) that are similar to observations. So it seems (I think) the authors are implying that the thermodynamical/dynamical responses of S4 to initial conditions may not eflect thermodynamical/dynamical changes induced by climate change. Again, I do not know why this would be (after all, climate models are used routinely for future projections, so they are assumed to work similarly in future as now). Please first explain how trends could impact predictability before drawing any conclusions on how climate change driven trends in data influence skill.

5) I am not very familiar with the VIC model, but I would assume that some of the internal states could be highly correlated/anticorrelated (as in most hydrological models). Using states averaged from different periods - as done here for the revESP method - could destroy these correlations, and could lead to unrealistically poor simulations. What I think the authors should have done is generated an ensemble of states by forcing VIC with resamples observations from a number of years (as in the original revESP) and then forced each of these with the ensemble of climate forecasts. This would lead to more ensemble members (15* number of years sampled), but would allow much more satisfacory diagnosis of the contribution of meteorological forcing to overall skill.

This means the revESP experiments would have more ensemble members than the other experiments, but I think the benefits of this approach outweigh potential artefacts arising from different ensemble sizes.

6) The method for the ESP is described as follows: "We did not select atmospheric forcings from observations (e.g. WFDEI), which is the strategy employed in most published ESP experiments. By selecting the forcing from the S4 hindcasts, the ESP experiments remain as close as possible to the Full Hindcasts." I assume the authors have used some way of sampling from different years, but the way this is written makes it possible to interpres this as if the ESP forecasts could simply be analogous to hindcasts. (If the authors have used 'ESP' in the latter way, 1) they should not call it ESP and 2) they cannot claim to isolate the source of skill, which is their aim.) The authors need to clarify what they have done here. If, as I've assumed, they have sampled from hindcasts, this is essentially a new method (which is a good thing). In this case the authors need to spell out their method clearly, including sampling strategies for ensemble members, etc.. They may also like to give it a new name - e.g. HESP for 'hindcast ESP', or similar.

7) As the authors are introducing a new system, it needs to be put in the context of existing operational and experimental prediction systems. The introduction does not really do this at present. For example, the authors could note that statistical systems can much more easily be configured to produce reliable ensemble forecasts (e.g. Madadgar and Moradkhani 2013; Wang and Robertson 2011). In addition, other experimental dynamical forecast systems have attempted to explicitly deal with problems related to reliability (Yuan 2016; Bennett et al. 2016). (While Yuan (2016) is mentioned, the authors do not note a crucial difference between this system and WHUSP - that is, Yuan's hydrological post-processing step that attempts to ensure reliable ensembles.) A paragraph explaining how the WHUSP approach compares to existing systems, including any differences in the aims of WHUSP compared to other systems, would be a useful addition to the introduction. (For example, it is fine to validate against pseudo-

reality if this is how the system is to be used in operation, but if the aim of WHUSP is to predict actual inflows to reservoirs then it should be validated against observations.)

8) Quantile mapping has several limitations as a method for post-processing ensemble forecasts - in particular that it does not correct for errors in reliability because it ignores information that is available from correlations between hindcasts and observations (see Wood and Schaake 2008; Madadgar et al. 2014). A statistical calibration (e.g. Gneiting et al. 2005; Schepen et al., 2014) is probably preferable. This should be acknowledged somewhere here.

9) It was not clear to me what was used as the reference forecast when RPSS was calculated. I suggest a cross-validated measure of climatology that varies with month (e.g. an ensemble of resampled historical streamflow, or similar). Please clarify.

Specific (minor) comments

Line 18 'The term ESP refers to...hindcasts'. Not only hindcasts - ESP systems are widely used to produce forecasts.

Line 20 'reference simulation'. As noted above, 'reference' forecasts in forecasting literature are frequently used to denote a benchmark for performance. I suggest using a different term than 'reference simulation' here, because it is not really being used as a reference. Suggest simply 'simulation'

Line 21 'identical'. ESP experiments are often cross-validated (depending on the aims of the experiment), so forcings may not be 'identical'.

Line 21 'This is not surprising'. It's not surprising if you consider correlation as synonymous with skill. As I argue above, I don't believe this is justified. As Murphy (1988) points out, skill scores have components that consider, e.g. conditional and uncon-

ditional biases, which correlations ignore. Clearly bias correction will influence these aspects of skill.

Lines 1-2 'By selecting the forcing from S4 hindcasts, the ESP experiments remain as close as possible to the Full Hindcasts'. I do not understand what the authors mean by 'select' here. It could imply that the authors simply used S4 forecasts (i.e. the same as the 'Full Hindcasts'), but this does not make sense (see comment #5). Please clarify.

Lines 14-15 '...which is important since ensemble size affects skill metrics'. I would say what's of more concern is that a small ensemble of 15 members is likely to mean that your skill metrics are subject to considerable sampling uncertainty.

Line 5 '...is linked to climate change...' change to '...could be linked to climate change...'

Line 5 '...by detrending the data...' A brief summary of what is being detrended, the detrending technique and the trend significance test is needed (either a description, which could go in an appendix, or a reference)

Line 6 '...is insignificant across most of the domain...' I assume the trends were analysed only for the hindcast periods. Please note this somewhere

Line 35 '(revESP) always causes much less significant skill' I accept that this will probably be true, but could this result be partly caused by the way in which that the model has been initialised? (i.e with states that may not be correlated - see comment #4)

Lines 1-3 'We explain the enhanced skill in runoff by an indirect effect of the skill of the precipitation forcing in the first lead month, which gradually adds some skill to the model states of soil moisture and snow.' I undestand what the authors are getting at here, but I think this is poorly phrased. The forcing doesn't really 'add skill' to the

model states. It's simpler to say that runoff forecasts are generally more skillful than climate forecasts because they aggregate skill from initial conditions and skill from meteorological forcings.

Line 31-32 '... where the first skill occurs...' What is meant by 'first skill'?

Line 20 'skill in evapotranspiration' - This section is somewhat out of character with the first part of the paper. I am not suggesting it is not important work, but it perhaps would have been better in its own paper.

Line 21 '...hindcasts of evapotranspiration have intrinsic value...' suggest '...hindcasts of evapotranspiration have value indepedent of streamflow forecasts...'. Also, please provide a brief summary somewhere of how VIC calculates ET.

P 10

Line 24-26 'Initial conditions of snow and/or soil conditions lead to skill in the temperature hindcasts of the climate model (S4) and initial conditions of snow and soil moisture lead to skill in the evapotranspiration hindcasts of the hydrological model (VIC).' Is there generally agreement in the VIC snow/soil moisture states and the (I presume) observations assimilated by S4?

P11

Line 18 '...their semi statistical forcing is more skilful than the S4 forcing...'. Is is also possible that your hydrological model is more efficient, thereby giving you relatively more skill from initial conditions?

Line 28 '...to what extent they are due to a lack of interannual variability in the processes that eliminate the skill?' As discussed in comment #1, this would be straightforward to answer if you used skill scores calculated against a suitable climatological reference forecast (comment #9)

Line 32 'which is an important skill-eliminating factor'. This won't show up in correlations.

P12

Line 25 'The logical answer is "yes" but such a strategy should then be reconsidered regularly'. As noted in comment #8, statistical calibration methods are available to post-process climate outputs. One of the benefits of these methods is that in the absence of demonstrable forecast skill, they return climatology forecasts. So a more effective alternative than using two forecasting systems might be to use calibrated S4 forecasts as forcing, which would then effectively give an ESP-like forecast when skill isn't there. See Peng et al. (2016) for an example of this applied to S4.

Line 36 'This study demonstrates the power of using pseudo-observations for verification.' I agree, but the usefulness of pseudo-reality also depends on the ultimate aims of the forecasting system. If the aim of the system is to give accurate streamflow forecasts where quantities matter (e.g., forecasting inflows to reservoirs), then the system must be verified accordingly (i.e., against gauged streamflows). I think the authors should acknowledge this.

Typos/grammar

Line 6 delete 'among others'

Line 37 '...of the an...' delete 'the'

Line 27-28 '...snow stops to be available...' change to 'snow is not available'

Line 34 delete 'quite'

Line 11 '...April as...' make '...April at...'

Line 30 change '...Mediterranean in due...' to '...Mediterranean is due...'

Line 5-6 '...two hotspot region.' Make it 'regions'

Line 33 '...not the case practical applications.' Add 'in'

References

Bennett, J. C., Q. J. Wang, M. Li, D. E. Robertson, and A. Schepen (2016), Reliable long-range ensemble streamflow forecasts: Combining calibrated climate forecasts with a conceptual runoff model and a staged error model, Water Resources Research, 52, 8238–8259, doi: 10.1002/2016wr019193.

Gneiting, T., A. E. Raftery, A. H. Westveld, and T. Goldman (2005), Calibrated probabilistic forecasting using ensemble model output statistics and minimum CRPS estimation, Monthly Weather Review, 133(5), 1098-1118, doi: 10.1175/mwr2904.1.

Madadgar, S., and H. Moradkhani (2013), A Bayesian Framework for Probabilistic Seasonal Drought Forecasting, Journal of Hydrometeorology, 14(6), 1685-1705, doi: 10.1175/jhm-d-13-010.1.

Madadgar, S., H. Moradkhani, and D. Garen (2014), Towards improved post-processing of hydrologic forecast ensembles, Hydrological Processes, 28(1), 104-122, doi: 10.1002/hyp.9562.

Murphy, A. H. (1988), Skill Scores Based on the Mean Square Error and Their Relationships to the Correlation Coefficient, Monthly Weather Review, 116(12), 2417-2424, doi: 10.1175/1520-0493(1988)116<2417:ssbotm>2.0.co;2.

Peng, Z., Q. J. Wang, J. C. Bennett, A. Schepen, F. Pappenberger, P. Pokhrel, and Z. Wang (2014), Statistical calibration and bridging of ECMWF System4 outputs for forecasting seasonal precipitation over China, Journal of Geophysical Research (Atmospheres), 119, 7116–7135, doi: 10.1002/2013JD021162.

Raftery, A. E. (2016), Use and communication of probabilistic forecasts, Statistical Analysis and Data Mining: The ASA Data Science Journal, 9(6), 397-410, doi: 10.1002/sam.11302.

Schepen, A., Q. J. Wang, and D. E. Robertson (2014), Seasonal forecasts of Australian rainfall through calibration and bridging of coupled GCM outputs, Monthly Weather Review, 142(5), 1758-1770, doi: 10.1175/mwr-d-13-00248.1.

Wang, Q. J., and D. E. Robertson (2011), Multisite probabilistic forecasting of seasonal flows for streams with zero value occurrences, Water Resources Research, 47, W02546, doi: 10.1029/2010WR009333.

Wood, A. W., and J. C. Schaake (2008), Correcting Errors in Streamflow Forecast Ensemble Mean and Spread, Journal of Hydrometeorology, 9(1), 132-148, doi: 10.1175/2007jhm862.1.

Yuan, X. (2016), An experimental seasonal hydrological forecasting system over the Yellow River basin – PartÂă2: The added value from climate forecast models, Hydrology and Earth System Sciences, 20(6), 2453-2466, doi: 10.5194/hess-20-2453-2016.

---

## Author Comment (AC3) · 14 Apr 2017

We are very happy with the fact that the editor found three anonymous referees to give their highly informed opinions on our paper. We thank all three for their respective efforts to produce such extensive and constructive reviews.

We adopt most of the suggested textual improvements and specified our action to every remark in the annotated report hess-2016-604-RC3-author-reply.pdf1. RC3 raises concerns regarding a number of technical issues (RC3's numbering):

3. Streamflow persistence due to e.g. aquifer fed flow (e.g. for he UK: Svesson et al., 2015 ERL) is at least partially accounted for through soil moisture stores. It is one of the objectives of the current study to analyse whether this store is sufficiently large to attain the observed levels of streamflow persistence. We will improve the introduction

and discussion sections on this aspect.

4. Wrt human impacted, regulated flows we concede that impacts of dams etc. on river flow have not been accounted for in our model nor in this paper. Thus, all results presented refer to naturalised flow. We will state that more explicitly. In paper I, a brief analysis has been made, partially attributing the reduction of skill between theoretical and actual to the effect of dams and other man-made channel modifications.

5. This relates to issue [4]. Actual skill here is defined against real station data, theoretical skill is measured against the reference simulation (pseudo observations, reanalysis data). This was extensively discussed in paper 1. However, we will make paper 2 more self-contained by restating such definitions and better explaining such statements, repeated from paper 1.

6. We (partially) address this issue in the discussion (fig 12 and ensuing texts). A more full exploration of these pure statistical relations between snow/soil moisture and streamflow will not inform us about the added value of using a model (since here they are model derived themselves), unless such analysis would be based on pure observational records of snow or soil moisture. That we consider beyond the scope of the present study.

7. Also raised by other RCs. We have the average R of all significant cells already available and will present them, probably as supplementary material.

8. Also RC2 made the same point. In paper 1 the other skill scores have been briefly presented, and its similarity in spatio-temporal patterns to those of R demonstrated. However we will better describe the differences observed between the various metrics. All metrics (R, RPSS, ROC AN, ROC BN) will be more fully presented in both the body text and in the supplementary material of paper 1. However, since paper 2 focuses on causes of skill, we believe it less relevant to show other metrics.

9. This part of the discussion focuses on the (potential) contribution of meteorological
forecast quality to stream flow forecast skill, showing that (partially) statistical meteo-rological forecasts seem to outperform those of purely dynamical state of art models, but that there is scope for improvement of the latter. A more extensive discussion on (potential progress in) meteorological forecast quality is beyond the scope of our paper (and of our expertise) . So we will try to explain better the implications of these findings, but not delve into meteorological forecast quality itself.

10. Assimilation of snow or soil moisture observations into our modelling framework is high on our wish list too, and will be future work. We can, however, try to better discuss its potential based on literature.

11. We recognize this and we will -also with help of suggestions by the other reviewers-make a serious attempt to improve and have a final check done by a native speaker.

Please also note the supplement to this comment:
http://www.hydrol-earth-syst-sci-discuss.net/hess-2016-604/hess-2016-604-AC3-supplement.pdf

**Supplement:**

The manuscript is the 2nd of a companion paper addressing streamflow forecast performance across Europe. The methodology is fairly well described and presented, however substantial clarification and justification is necessary in numerous areas. The authors predominantly limit themselves to evaluation of only a few performance metrics, and report many findings for the whole of Europe. The overall contribution contains meritorious aspects, particularly the performance of this dynamical system, however these need to be highlighted and clarified significantly. Also, distinction and improvement between this and prior studies (e.g. Bierkens and van Beek) is not sufficient.

Specific comments: 1. The title indicates seasonal streamflow prediction, yet the paper focuses on Monthly results for streamflow, temperature, and evaporation. Title not entirely indicative of manuscript focus.

2. Given the spatial heterogeneity of Europe, the authors should provide better justification for reporting predominantly spatially lumped results.

3. Auto-regressive effect (streamflow persistence) not explicitly mentioned or discussed. Is it assumed to be (partially) accounted for in initial soil moisture? For most rivers, particularly large rivers, this is a dominant feature.

4. GRDC has discharge stations downstream of reservoirs, where regulations and management of discharge is often evident. But these have not been corrected (or even noted) in the dataset. Europe is full of situations like this. How have those been accounted for?

5. Unclear (no explanation) of what the ratio of actual/theoretical skill means. Clear that they are closer for larger basins (no surprise) but does the fact that both are far from 1 indicate less "realistic" outcomes? Or does this have little bearing on skill metrics (comparing apples to apples.) Please clarify.

6. How do the three "conditions" relate with the pseudo-obs? For example, report R or RPSS between soil moisture and streamflow by grid? Or snow and streamflow? Could assess for at least a sub-set of locations. This would also give insights as to the value added (or not) by VIC.

7. In addition to reporting the % of cells where R is significant, consider also reporting the mean and standard deviation of R in those cells. The number of significant cells does not necessarily represent the quality of the relationship (e.g. % of cells could increase, but mean decrease. . .) And then discuss.

8. RPSS is mentioned early in the study, but results are not presented. Such categorical skill scores are worth exploring.

9. The authors lightly compare their study outputs with others, namely Bierkens and van Beek, indicating lower performance, likely attributable to the latter's use of semi-statistical forcing. While there are still other meritorious aspects to this current contribution, the authors do not adequately discuss the implications of poorer performance. Are there reasons that the proposed methodology is advantageous as compared with others? Should the GloSea5 approach be used in lieu of the one proposed here? More discussion is needed.

10. In the Conclusion, the authors mention the potential improvement of assimilating soil moisture or SWE to VIC. Why was this not performed and analyzed?

11. Challenging to follow train of through in some parts. Could benefit from the writing be tightened up overall - and simplified in some places. Word choice also needs to be improved in many places (e.g. "Fig. 8 analyses [sic] a remarkable feature." Figures cannot analyze. Figures can illustrate.)

---

## Referee Report (RR1)

**Review of "Seasonal streamflow forecasts for Europe – II. Sources of skill" by Wouter Greuell et al.**

This paper looks at the sources of skill in the WUSHP model-based seasonal hydrological forecasting system, producing hydrological forecasts for up to seven months of lead time over Europe. To this end, the authors first analysed the skill of the meteorological forcings of WUSHP (ECMWF's System 4 seasonal meteorological hindcasts). They also produced several hindcast datasets, based on which they carried out a complete investigation of the sources of skill (i.e. initial conditions of soil moisture and snow and meteorological forcings) in the seasonal runoff, discharge and evapotranspiration hindcasts.

The results presented in this paper are based on a thorough and original analysis and provide a great contribution to the field's existing literature. Furthermore, this paper is overall coherently written and I recommend it to be accepted after minor revisions. Below, a few comments which should hopefully guide the authors in revising this paper for publication.

**Main comments:**

- Since the last version of this paper, the authors do not seem to have updated their references list with the latest literature within the field, more specifically the papers published in the same HESS special issue on "Sub-seasonal to seasonal hydrological forecasting". This is for instance apparent on P26 L841-844, where the authors mention that to their knowledge, only two other studies looked at the sources of skill in seasonal hydrological forecasts produced by a similar system, over Europe. There are several papers in the same special issue which look at the skill of (System 4-driven) seasonal hydrological forecasts over the globe, Europe, or parts of Europe. These papers all benchmark the skill of the state-of-the-art seasonal hydrological forecasts to forecasts such as the ESP. I would highly recommend for the authors to update their references with this latest literature.

- This paper is very rich in results and figures. I would recommend for the authors to present the results in a more concise manner when possible and to remove non-essential figures from the main body. This would help highlight the key results of this paper (in my opinion and according to the paper's title, the analysis of the origins of skill in the runoff hindcasts) and keep the readers' focus throughout the paper. I have made below a few more specific comments about this.

- The legend is too small for most of the sub-figures.

**Title:** in my opinion, the title doesn't fully reflect the content of the paper. Firstly, this paper looks at runoff and discharge and the word "streamflow" is not used much in the paper. Furthermore, this paper presents much more than just a streamflow analysis. I would therefore recommend changing the title to something more representative of the content of the paper, for example: "Seasonal hydro-meteorological forecasts for Europe: sources of skill".

**Abstract:** the abstract is overall too long and dives into the results in great detail before explaining the methods used. I would recommend laying out the methods used more clearly before mentioning the results. The results' paragraph should be much shorter and highlight the key results of the paper, leaving the details for the main body. You use the word "streamflow" here and on a few occasions in the paper (e.g. P30 L973) while the rest of the time the words "runoff" or "discharge" are used. Please consider removing the word "streamflow" from the paper or clarifying what it refers to.

**P4 L95-96:** please specify here which variables this analysis is based on.

**P4 L109-110:** you need to explain here what the ESP is for readers not yet familiar with it.

**P4 L109-116:** it would be good to explain more clearly here that the hindcasts you produced for this paper are inspired from the conventional ESP and reverse-ESP, but that they differ in their set up and why that is (with regards to the overall aim of the paper).

**P4 L118:** could you please reference here one or multiple papers that have looked at the effect of evapotranspiration on runoff, more specifically on seasonal timescales.

**P5 L165-166:** "a length of seven months" is ambiguous. Using the word lead time would be clearer.

**P6 L195-196:** do the observations you mention here refer to the pseudo-observations?

**P7 L235-236:** this explanation of the target month number is slightly confusing and not necessarily needed. You could simply say that the target month refers to the month for which the forecast is made. I also find jumping between the terms "lead month" and "lead time" throughout the paper a bit confusing.

**P7 L245-251:** please remind the reader that in this paper you use alternative methods to the standard ESP and reverse-ESP (widely used in the literature) because you want to keep these hindcasts as close as possible to the FullSH for the aim of this paper.

**P8 L252-271:** I like the experiments' new names (after the first revision of the paper). Does snow conditions refer to the snow cover?

**P8 L274:** do you mean "random" instead of "uneven"?

**P9 L308:** please clarify what you mean by "most important input variables". Most important for what?

**3.1:** did you look at the effect of trends in the System 4 precipitation hindcasts?

**Figure 1:** in the caption, it says that "yellow cells have insignificant skill". There are however multiple shades of yellow on the map. You could instead say that the lightest yellow colour shows insignificant skill. P10 L328-329: for which lead months?

**P12 L372-373:** the lead month 0 line for the un-detrended hindcasts looks very close to the line for the detrended hindcasts, except for April, June and November.

**P12 L373-374:** it however doesn't drop as quickly as for precipitation.

**P12 L395-396:** I agree with this observation for JJA, while for the other months it is not so evident.

**P13 L415:** why was lead month 5 selected to illustrate this point? Do the other lead months show (dis)similar results?

**P13 L421-423:** for the sake of conciseness, I would maybe remove these results.

**P13 L428-429:** could this be due to some trend or predictor that takes shape in autumn and affects temperatures in Europe in winter? I wouldn't dismiss it as spurious.

**P14 L472-474:** the difference is however not very significant, except in late summer-autumn.

**Figure 4:** I would suggest moving the bottom two sub-figures to the supplementary material as it won't affect the main storyline of the results.

**Figure 5:** green and red should never be used together on a plot. A more colourblind friendly palette should be used instead (same for Figure 13). Since you mention the FullSH in the results that correspond to this figure it would make sense to add a line for the FullSH here as well.

**P17 L537-547:** these are great results!

**P17 L549-550:** could that be because the snowpack is at its maximum around February in Europe?

**P17 L565-569:** I wouldn't mention these results here unless you explain what the differences could be due to.

**Figure 6:** is this figure essential in the main body? You only discuss the additivity of skill from the initial soil moisture and snow conditions, which Figure 5 already shows as an average over Europe. In my opinion, the text is sufficient here.

**P18 L583-585:** this sentence is slightly confusing.

**P19-20 L619-620:** could this be rather due to the groundwater initialisation?

**Figure 8:** this figure is not necessary and a few sentences are sufficient to raise this point.

**P20-21 L637-653:** couldn't this be due to the fact that we can expect most of the snow in Europe (except for high mountain ranges) to have melted already by May? Knowing the snow cover at the start of May is therefore of not much added value for forecasting future runoff compared to knowing what it is earlier in the year, in April for example (when more snow is still present and available for runoff).

**3.3:** this part is too long and steals the spotlight from the runoff section, which should be the highlight of the paper according to the title. I would therefore suggest to summarise the text corresponding to Figures 10 and 11 in just a few lines and remove these figures from the main body.

**P22 L695-698:** I would describe what is observed on sub-figure 9b before comparing those results to the ones obtained for runoff.

**P22 L700-703:** please mention that these results correspond to the FullSH.

**P26 L819-822:** many studies (published in the same special issue) found higher skill in the ESP (compared to a state-of-the-art seasonal hydrological forecasting system) beyond the first or second month of lead time over Europe or for specific basins/regions of Europe and are worth mentioning here.

**P29 L935:** I am not sure to understand what is meant by "inter-member variability".

**P29 L942-944:** one further difference is that the initial conditions in the reverse-ESP are the full range (or ensemble) of historical initial conditions, instead of a single value (i.e. climatological average).

**P30 L953-954:** this is a repetition of what was said in the previous paragraph.

**P31 L1007-1013:** this is a very good point!

**P33 L1080-1081:** I would remove this example from the conclusions.

**Technical corrections:**

- P14 L445: lead month 1 instead of 0?

- P16 L524: "run-off" is used instead of "runoff" (comes up again after).

- P22 L686: the "o" is missing in "MeteoSH".

- P27 L879: "taken from" instead of "taking from".

---

## Author Response (AR2)

**Dear Editor**

Here below is the response to your own suggestions. All our replies to the suggestions by the reviewers are added as comments to the documents with their suggestions.

**References**

I updated the literature by adding sentences referring to the work by Arnal et al. (2018), Harrigan et al. (2018), Meissner et al. (2017) and Bazile et al. (2017).

**Paper length:**

Reviewer 2 put forward three suggestions for shortening the paper. We believe that the evaluation of skill in evapotranspiration is important, as argued in the beginning of Section 3.3, and fits very well into the present paper. So, we did not remove it.

The second suggestion was to cut on Section 3.4. This section was added to the paper of the second submission is response to suggestions by reviewer 1, e.g. his main point 2:

*please consider either arguing more thoroughly why you have decided to take a different approach from the standard ESP and reverse-ESP or changing the experiment designs.*

And in response to suggestions by the editor, e.g.

*Please clarify or repeat the experiment, or even clearly state in the objectives and conclusions the different aims of your approach in relevance to the original ESP/revESP.*

Perhaps, we somewhat overreacted, so Section 3.4 became too long. I shortened the section considerably (from 870 to 664 words) and moved the figure to the supplementary material.

We followed the third suggestion of Reviewer 3 and removed Figure B2. It is now in the supplementary material).

In total, we removed 2.5 figures, namely Figure 13, Figure B2 and half of Figure 4. I also shortened the paper by removing words, sentences and paragraphs. I removed a paragraph from the introduction and two paragraphs from Section 4.4 (Towards an operational system)

**Reliability**

We acknowledge that reliability is an important property of forecasts and that it is hence valuable to publish an analysis of the reliability of the hindcasts. However, the topic of the present paper is an **explanation of the skill**, and not the reliability, of our hindcasts. So, the reliability evaluation that we added in an appendix is only vaguely related to the contents of the paper and **would seriously break the flow of the paper** if it was included in the main text. In the companion paper, however, **skill was analysed**. Hence, analysis of reliability would have well fitted in the first paper but we are obviously too late to do so. We see two reasonable solutions: 1) publish an addenda to the companion paper or 2) leave the evaluation of reliability in the appendix of the current paper, and summarise the analysis and refer to the appendix in the main text. In the new version we have implemented the last option, with the following text in the introduction of the paper:

*To extend the evaluation of the system, its reliability was analysed. The main finding is that during the two first lead months the system is not far from being perfectly reliable but that with progressing lead time reliability is reduced. We also found that discrimination skill and reliability have similar characteristics, e.g. for longer lead times the highest values of reliability are found in some regions with considerable amounts of discrimination skill. Details of this analysis are provided in Appendix A.*

**Review of "Seasonal streamflow forecasts for Europe – II. Sources of skill" by Wouter Greuell et al.**

This paper looks at the sources of skill in the WUSHP model-based seasonal hydrological forecasting system, producing hydrological forecasts for up to seven months of lead time over Europe. To this end, the authors first analysed the skill of the meteorological forcings of WUSHP (ECMWF's System 4 seasonal meteorological hindcasts). They also produced several hindcast datasets, based on which they carried out a complete investigation of the sources of skill (i.e. initial conditions of soil moisture and snow and meteorological forcings) in the seasonal runoff, discharge and evapotranspiration hindcasts.

The results presented in this paper are based on a thorough and original analysis and provide a great contribution to the field's existing literature. Furthermore, this paper is overall coherently written and I recommend it to be accepted after minor revisions. Below, a few comments which should hopefully guide the authors in revising this paper for publication.

**Main comments:**

- Since the last version of this paper, the authors do not seem to have updated their references list with the latest literature within the field, more specifically the papers published in the same HESS special issue on "Sub-seasonal to seasonal hydrological forecasting". This is for instance apparent on P26 L841-844, where the authors mention that to their knowledge, only two other studies looked at the sources of skill in seasonal hydrological forecasts produced by a similar system, over Europe. There are several papers in the same special issue which look at the skill of (System 4-driven) seasonal hydrological forecasts over the globe, Europe, or parts of Europe. These papers all benchmark the skill of the state-of-the-art seasonal hydrological forecasts to forecasts such as the ESP. I would highly recommend for the authors to update their references with this latest literature.

- This paper is very rich in results and figures. I would recommend for the authors to present the results in a more concise manner when possible and to remove non-essential figures from the main body. This would help highlight the key results of this paper (in my opinion and according to the paper's title, the analysis of the origins of skill in the runoff hindcasts) and keep the readers' focus throughout the paper. I have made below a few more specific comments about this.

- The legend is too small for most of the sub-figures

**Title:** in my opinion, the title doesn't fully reflect the content of the paper. Firstly, this paper looks at runoff and discharge and the word "streamflow" is not used much in the paper. Furthermore, this paper presents much more than just a streamflow analysis. I would therefore recommend changing the title to something more representative of the content of the paper, for example: "Seasonal hydro-meteorological forecasts for Europe: sources of skill".

**Abstract:** the abstract is overall too long and dives into the results in great detail before explaining the methods used. I would recommend laying out the methods used more clearly before mentioning the results. The results' paragraph should be much shorter and highlight the key results of the paper, leaving the details for the main body. You use the word "streamflow" here and on a few occasions in the paper (e.g. P30 L973) while the rest of the time the words "runoff" or "discharge" are used. Please consider removing the word "streamflow" from the paper or clarifying what it refers to.

**P4 L95-96:** please specify here which variables this analysis is based on.

**P4 L109-110:** you need to explain here what the ESP is for readers not yet familiar with it.

**P4 L109-116:** it would be good to explain more clearly here that the hindcasts you produced for this paper are inspired from the conventional ESP and reverse-ESP, but that they differ in their set up and why that is (with regards to the overall aim of the paper).

**P4 L118:** could you please reference here one or multiple papers that have looked at the effect of evapotranspiration on runoff, more specifically on seasonal timescales.

**P5 L165-166:** "a length of seven months" is ambiguous. Using the word lead time would be clearer.

**P6 L195-196:** do the observations you mention here refer to the pseudo-observations?

**P7 L235-236:** this explanation of the target month number is slightly confusing and not necessarily needed. You could simply say that the target month refers to the month for which the forecast is made. I also find jumping between the terms "lead month" and "lead time" throughout the paper a bit confusing.

**P7 L245-251:** please remind the reader that in this paper you use alternative methods to the standard ESP and reverse-ESP (widely used in the literature) because you want to keep these hindcasts as close as possible to the FullSH for the aim of this paper.

**P8 L252-271:** I like the experiments' new names (after the first revision of the paper). Does snow conditions refer to the snow cover?

**P8 L274:** do you mean "random" instead of "uneven"?

**P9 L308:** please clarify what you mean by "most important input variables". Most important for what?

**3.1:** did you look at the effect of trends in the System 4 precipitation hindcasts?

**Figure 1:** in the caption, it says that "yellow cells have insignificant skill". There are however multiple shades of yellow on the map. You could instead say that the lightest yellow colour shows insignificant skill. P10 L328-329: for which lead months?

**P12 L372-373:** the lead month 0 line for the un-detrended hindcasts looks very close to the line for the detrended hindcasts, except for April, June and November.

**P12 L373-374:** it however doesn't drop as quickly as for precipitation

**P12 L395-396:** I agree with this observation for JJA, while for the other months it is not so evident

**P13 L415:** why was lead month 5 selected to illustrate this point? Do the other lead months show (dis)similar results?

**P13 L421-423:** for the sake of conciseness, I would maybe remove these results.

[Figure]

**P13 L428-429:** could this be due to some trend or predictor that takes shape in autumn and affects temperatures in Europe in winter? I wouldn't dismiss it as spurious.
[Figure]

**P14 L472-474:** the difference is however not very significant, except in late summer-autumn.

**Figure 4:** I would suggest moving the bottom two sub-figures to the supplementary material as it won't affect the main storyline of the results.

**Figure 5:** green and red should never be used together on a plot. A more colourblind friendly palette should be used instead (same for Figure 13). Since you mention the FullSH in the results that correspond to this figure it would make sense to add a line for the FullSH here as well.

**P17 L537-547:** these are great results!

**P17 L549-550:** could that be because the snowpack is at its maximum around February in Europe?

**P17 L565-569:** I wouldn't mention these results here unless you explain what the differences could be due to.

**Figure 6:** is this figure essential in the main body? You only discuss the additivity of skill from the initial soil moisture and snow conditions, which Figure 5 already shows as an average over Europe. In my opinion, the text is sufficient here.

**P18 L583-585:** this sentence is slightly confusing.

**P19-20 L619-620:** could this be rather due to the groundwater initialisation?

**Figure 8:** this figure is not necessary and a few sentences are sufficient to raise this point.

**P20-21 L637-653:** couldn't this be due to the fact that we can expect most of the snow in Europe (except for high mountain ranges) to have melted already by May? Knowing the snow cover at the start of May is therefore of not much added value for forecasting future runoff compared to knowing what it is earlier in the year, in April for example (when more snow is still present and available for runoff).

**3.3:** this part is too long and steals the spotlight from the runoff section, which should be the highlight of the paper according to the title. I would therefore suggest to summarise the text corresponding to Figures 10 and 11 in just a few lines and remove these figures from the main body

**P22 L695-698:** I would describe what is observed on sub-figure 9b before comparing those results to the ones obtained for runoff.

**P22 L700-703:** please mention that these results correspond to the FullSH.

**P26 L819-822:** many studies (published in the same special issue) found higher skill in the ESP (compared to a state-of-the-art seasonal hydrological forecasting system) beyond the first or second month of lead time over Europe or for specific basins/regions of Europe and are worth mentioning here.

**P29 L935:** I am not sure to understand what is meant by "inter-member variability".

**P29 L942-944:** one further difference is that the initial conditions in the reverse-ESP are the full range (or ensemble) of historical initial conditions, instead of a single value (i.e. climatological average).

**P30 L953-954:** this is a repetition of what was said in the previous paragraph.

**P31 L1007-1013:** this is a very good point!

**P33 L1080-1081:** I would remove this example from the conclusions

**Technical corrections:**

- P14 L445: lead month 1 instead of 0?

- P16 L524: "run-off" is used instead of "runoff" (comes up again after)

- P22 L686: the "o" is missing in "MeteoSH"

- P27 L879: "taken from" instead of "taking from".

**Report[GW1] #2**

Submitted on 22 Oct 2018
Anonymous Referee #2

**Anonymous during peer-review: Yes** No

**Anonymous in acknowledgements of published article: Yes** No

**Recommendation to the Editor**

| | |
|---|---|
| **1) Scientific Significance**
Does the manuscript represent a substantial contribution to scientific progress within the scope of this journal (substantial new concepts, ideas, methods, or data)? | Excellent **Good** Fair Poor |
| **2) Scientific Quality**
Are the scientific approach and applied methods valid? Are the results discussed in an appropriate and balanced way (consideration of related work, including appropriate references)? | Excellent Good **Fair** Poor |
| **3) Presentation Quality**
Are the scientific results and conclusions presented in a clear, concise, and well structured way (number and quality of figures/tables, appropriate use of English language)? | Excellent **Good** Fair Poor |

For final publication, the manuscript should be

**accepted as is**

accepted subject to **technical corrections**

**accepted subject to minor revisions**

reconsidered after **major revisions**

    I am willing to review the revised paper.

    I am **not** willing to review the revised paper.

**rejected**

**Suggestions for revision or reasons for rejection (will be published if the paper is accepted for final publication)**

Major comments[GW2]

This is a resubmitted version of a study describing the sources of skill in a Europe wide streamflow forecasting system. It's my second review of the paper. As I noted last time, dynamical continental scale ensemble prediction systems are at the cutting edge of seasonal streamflow forecasting, the paper is reasonably well presented, and it is well within the scope of HESS. In addition, the authors **have addressed several of my concerns, including improving the description of their analyses of forecast skill and trends, and peforming analyses of reliability.** They have also added some interesting analyses of sources of skill in **Figure 12**. I commend them on their efforts. **There are still some outstanding issues**, however. Some are repetitions of my statements last time, others have arisen from the revision. They are as follows:

1) **Reliability**: the authors have expended considerable effort on investigating reliability with attributes diagrams. Unfortunately, they have placed all this effort in an **Appendix**, and have not referred to that appendix at all in the body of the paper. As I stated previously: reliability is a crucial attribute of ensemble forecasting systems, and merits discussion. This is particularly true when the main analyses used in this paper reduces the ensemble to a deterministic forecast (i.e., correlations rather than the use of probabilistic scores). I recommend at least some of this analysis be moved into the body of the paper. The reliability results are not particularly strong - the ensembles appear strongly overconfident at longer lead times - but I think this is an avenue for future improvement. (NB - see also my suggestions for shortening the paper, below.)

2) Paper Length: the addition of analyses and discussion has resulted in a **paper** that is, in my view**, too long**. I offer three possible ways to shorten the paper:
i) I reiterate my recommendation from my previous review that the **analyses of evapotranspiration forecasts** be removed from this paper, and given its own paper.
ii) Figure 13, and its accompanying discussion, is superficial and could easily be removed. One of the major benefits of ESP forcings is that they offer a reliable estimate of uncertainty. This is distinct from the authors' InitSH experiment, which samples from Sys4 (resulting in an ensemble that is likley to be overconfident). This of course does not show up in correlations calculated on the median ensemble member. **The authors could simply state something like "The use of InitSH produced very similar correlations to ESP (not shown for brevity)."** If the authors feel strongly that Fig 13 should be included, they could put it in as another panel in Figure 5 (and reduce the discussion of it to a sentence or two), but I think this is unnecessary.
iii) The **analysis of reliability of actual streamflow forecasts (Figure B2) is probably unnecessary**, as the remainder of the paper verifies against pseudo observations. It could be removed.

Specific (minor) comments

L101 "specific hindcasts" - would 'experiments' perhaps be a better term[GW3]?

L125-132 Arnal et al. 2018 has done this comprehensively over Europe recently, and is worth mentioning, both here and in the discussion of your results[GW4].

L175 "To spin up discharge, each 7-month hindcast was preceded by a one month simulation" The companion paper implies that the hydrological states at the start of the 1-month spin-up are taken from a long-run simulation (if I've interpreted this correctly). This is important information (!) and should be included. At present, it reads as though only a single month is used to spin-up hydrological model states, which is nowhere near enough[GW5].

L196 "from the observations themselves" Should this be 'pseudo-observations[GW6]'?

L252-274 This information is perhaps better presented in a table, for easy reference[GW7].

L256-257 "More specifically, we selected member 1 from the 1981 hindcasts, member 2 from the 1983 hindcasts, etc. By using identical meteorological forcing for all of the years of the hindcasts, skill due to skill in the forcing is eliminated." Does this effectively mean that an ensemble member is drawn randomly from each year[GW8]?

L270-271 "skill due to initial conditions is eliminated" Should that be "skill due to initial hydrological conditions is eliminated[GW9]"?

L274 "15 uneven years[GW10] 1981-2009" I take it this means the years {1981, 1983, ..., 2007, 2009}(?) Does this mean the exact (perfect) rainfall forecast is included in the ensemble when assessing odd years? I.e., if you are evaluating forecasts for the year 2009, the observed rainfall for that year is included in the forecast ensemble[GW11]?

L288-289 "Thus, like the FullSH, all specific hindcasts for a single starting date consist of 15 members, which is important since ensemble size affects skill metrics". Of far greater concern is strict cross-validation of forecasts. Including a 'perfect' rainfall forecast in an ensemble of 15 is likely to have a much greater impact on skill scores than ensemble size. I don't think this is defensible. (Though as I recommend removing the ESP figures/experiment, above, it does not need to be addressed[GW12].)

L401-420 I think the additons to this explanation of what you're trying to show with your analysis of temperature trends has improved it considerably - it's much clearer to me (and hopefully others) now. There are a few instances later in the paper where phrases such as[GW13] "

L500 "The InitSH forcing is the same for each year, so its interannual variation does not contain a signal nor noise." The 'signal' depends on observations, which vary from year to year. So you can't say there is no 'noise' - the accuracy of InitSH changes each year, because the observations vary. Rather than centering this discussion on signal and noise, it would be easier to simply talk about forecast accuracy. I think the main conclusion to draw from Fig 4 is that when the WHUSP forecasts are reduced to the median, at longer lead times the meteorological forecasts from FullSH are less accurate than the randomly drawn met forecasts in InitSH. This is possible, of course, and a reason why a number of studies choose calibration methods rather than simple bias-corrections to process meteorological forecasts (see Zhao et al. 2017) (and also one of the reasons why ESP forecasting systems are difficult to beat[GW14]).

L645-646 "However, there is compensation for this direct effect by an indirect effect through soil moisture." Is it possible that this is an artefact of breaking correlations between states? i.e., SnInitSH has averaged soil moisture states at lead 0 - could running the model induce more correctly correlated soil moisture states at lead 1? In other words, could this be an artefact of your choice to average states, rather than using an ensemble of model states as standard revESP experiments do? If so, I think this should be acknowledged somewhere[GW15].

L542-L543 "Skill in the precipitation forcing of the first lead month leads to skill in the states of soil moisture and snow at the end of that month." I would guess skillful forecasts occur at long lead times in at least some catchments where there is no skill in precipitation in the first month. Correctly initialised hydrological models can produce skillful streamflow forecasts for a number of months, even with completely uninformative forcings[GW16].

L817-819 "This result is counter-intuitive but, as we discussed, a logical consequence of forcing with interannual variation
that has no or insufficient skill, such as the S4 forcing[GW17]." This doesn't sound logical to me at all. InitSH has, by design, zero meteorological forecast skill - so you cannot explain the poor S4 performance by saying it S4 has no meteorological forcing skill. (If this were so, it should perform similarly to InitSH, not worse than it.) I think the explanations possible are: 1)there are actual flaws in the forecast S4 forcings; this might be because of bias (though this is unlikely, as I'm sure the quantile mapping takes care of this) or that the S4 forecasts are negatively skillful (i.e., less accurate than and ESP forecast - this looks the most likely candidae) at longer lead times. 2) the way you've assessed the forecast insn't sufficiently sensitive to determine

L821-822 "have identical meteorological forcing for each year". I think I commented on this in the past revision: ESP forecasts are frequently not identical for each year, because they are often cross-validated. So they are similar, but not identical. They are similar to InitSH in that they are (or should be) uninformative, but give a reliable uncertainty spread[GW18].

L841 "To our knowledge, two other studies analysed" as noted previously, Arnal et al. 2018 have done very similar work to that presented here, and with the same forcing[GW19].

L859 "In any case, that contribution depends and will depend on the climate model used (e.g. S4 or GloSea5)." It can also very much depend on the method used to process climate forecasts. Calibration removes bias, but also ensures consistently inaccurate forecasts (i.e., negatively skillful forecasts, in the sense that they perform worse than climatological reference forecasts) return to something like a climatology forecast. This allows skill at short lead times in meteorological forecasts to propagagate through to long lead times in streamflow predictions. Quantile mapping only removes bias[GW20].

L879 "taking" should be "taken[GW21]"

L883 Figure 12 - headings say 'as lead' when they should say 'at lead[GW22]'

l1031 "this is not the case for practical applications" - well, it probably says that if you are choosing a benchmark/reference forecast, climatology is probably not good enough; it would be more stringent to use a benchmark of climatology+trend[GW23][GW24].

L1174 Figure B1 - This is a nice figure, but wouldn't it have been better to look at forecasts exceeding the median, to make it more consistent with the calculation of correlations (which are calculated on the median ensemble member[GW25])?

Typos/Grammar

L145 "So, the objective" Delete "So, " [GW26]

L205 "and to be available" should be "and are available[GW27]"

L675 Delete the first 'because[GW28]'.

[revised manuscript text omitted]

~~Figure 13 shows results for three different lead times. In the graph most of the points for the ESP are indistinguishable from their counterparts for the InitSH. So, all conclusions that were drawn from Fig. 4 and especially the reversal with lead time of the ranking of predictability for the FullSH and the InitSH, are equally true for the ranking of the FullSH and the ESP. We also conclude that, though forcings in the InitSH and the ESP differ, skills from both types of specific hindcasts are, as expected, virtually identical, which can be ascribed to the fact that the forcings do not vary from year to year. We speculate that this result would also hold for other plausible forcings that do not vary from year to year.~~

This  similarity of the InitSH and ESP is in sharp contrast with the skill resulting from reverse-ESP and MeteoSH, which are expected to be totally different. Keeping in mind that in both types of hindcasts skill is caused only by skill of the meteorological forcing, this is the skill of the S4 hindcasts in the MeteoSH. The present study showed that in Europe there is a small contribution to skill in the  runoff hindcasts by the forcing and that this contribution tends to decrease with time. This differs from reverse-ESP, in which skill is small at the beginning and then increases with lead time to reach perfect skill at very long leads (see Wood and Lettenmaier, 2008) because the meteorological forcing is quasi-perfect (i.e. identical to the forcing in the reference simulation) while the influence of the initial conditions, which are non-informative in reverse-ESP , decreases with time.

~~In summary, amounts of skill are almost the same for ESP and InitSH, and totally different for reverse-ESP and MeteoSH. Also, interpretations of reverse-ESP and MeteoSH differ. MeteoSH can be used to assess skill in the streamflow hindcasts due exclusively to skill in the meteorological hindcasts. Reverse-ESP can be used to quantify skill due to prescribing meteorological observations, i.e. the skill if we had perfect~~

[revised manuscript text omitted]